# Matrix Completion with Quantified Uncertainty through Low Rank Gaussian Copula

**Yuxuan Zhao**
Cornell University
yz2295@cornell.edu

**Madeleine Udell**
Cornell University
udell@cornell.edu

## Abstract

Modern large scale datasets are often plagued with missing entries. For tabular data with missing values, a flurry of imputation algorithms solve for a complete matrix which minimizes some penalized reconstruction error. However, almost none of them can estimate the uncertainty of its imputations. This paper proposes a probabilistic and scalable framework for missing value imputation with quantified uncertainty. Our model, the Low Rank Gaussian Copula, augments a standard probabilistic model, Probabilistic Principal Component Analysis, with marginal transformations for each column that allow the model to better match the distribution of the data. It naturally handles Boolean, ordinal, and real-valued observations and quantifies the uncertainty in each imputation. The time required to fit the model scales linearly with the number of rows and the number of columns in the dataset. Empirical results show the method yields state-of-the-art imputation accuracy across a wide range of data types, including those with high rank. Our uncertainty measure predicts imputation error well: entries with lower uncertainty do have lower imputation error (on average). Moreover, for real-valued data, the resulting confidence intervals are well-calibrated.

## 1 Introduction

Missing data imputation forms the first critical step of many data analysis pipelines; indeed, in the context of recommender systems, imputation itself is the task. The remarkable progress in low rank matrix completion (LRMC) [6, 25, 36] has led to wide use in collaborative filtering [37], transductive learning [17], automated machine learning [43], and beyond. Nevertheless, reliable decision making requires one more step: assessing the uncertainty of the imputed entries. While multiple imputation [39, 28] is a classical tool to quantify uncertainty, its computation is often expensive and limits the use on large datasets. For single imputation methods such as LRMC, very little work has sought to quantify imputation uncertainty. The major difficulty in quantifying uncertainty lies in characterizing how the imputations depend on the observations through the solution to a nonsmooth optimization problem. Chen et al. [9] avoids this difficulty, providing confidence intervals for imputed real valued matrices, by assuming isotropic Gaussian noise and a large signal-to-noise ratio (SNR). However, these assumptions are hardly satisfied for most noisy real data.

The probabilistic principal component analysis (PPCA) model [41] provides a different approach to quantify uncertainty. The PPCA model posits that the data in each row is sampled iid from a Gaussian factor model. In this framework, each missing entry has a closed form distribution conditional on the observations. The conditional mean, which is simply a linear transformation of the observations, is used for imputation [44, 22]. However, the Gaussian assumption is unrealistic for most real datasets.

The Gaussian copula model presents a compelling alternative that enjoys the analytical benefits of Gaussians and yet fits real datasets well. The Gaussian copula (or equivalently nonparanormal distribution) [29, 12, 15, 21] can model real-valued, ordinal and Boolean data by transforming a latent

Gaussian vector to match given marginal distributions. Recently, Zhao and Udell [45] proposed an imputation framework based on the Gaussian copula model and empirically demonstrated state-of-the-art performance of Gaussian copula imputations on long skinny datasets. However, their algorithm scales cubically in the number of columns, which is too expensive for applications to large-scale datasets such as collaborative filtering and medical informatics.

**Our contribution**   We propose a low rank Gaussian copula model for imputation with quantified uncertainty. The proposed model combines the advantages of PPCA and Gaussian copula: the probabilistic description of missing entries allows for uncertainty quantification; the low rank structure allows for efficient estimation from large-scale data; and the copula framework provides the generality to accurately fit real-world data. The imputation proceeds in two steps: first we fit the LRGC model, and then we compute the distribution of the missing values separately for each row, conditional on the observed values in that row. We impute the missing values with the conditional mean and quantify their uncertainty with the conditional variance. Our contributions are as follows.

1. We propose a probabilistic imputation method based on the low rank Gaussian copula model to impute real-valued, ordinal and Boolean data. The rank of the model is the only tuning parameter.
2. We propose an algorithm to fit the proposed model that scales linearly in the number of rows and the number of columns. Empirical results show our imputations provide state-of-the-art accuracy across a wide range of data types, including those with high rank.
3. We characterize how the mean squared error (MSE) of our imputations depends on the SNR. In particular, we show the MSE converges exponentially to the noise level in the limit of high SNR.
4. We quantify the uncertainty of our estimates. Concretely, we construct confidence intervals for imputed real values and provide lower bounds on the probability of correct prediction for imputed ordinal values. Empirical results show our confidence intervals are well-calibrated and our uncertainty measure predicts imputation error well: entries with lower estimated uncertainty do have lower imputation error (on average).

**Related work**   Although our proposed model has a low rank structure, it greatly differs from LRMC in that the observations are assumed to be generated from, but not equal to, a real-valued low rank matrix. Many authors have considered generalizations of LRMC beyond real-valued low rank observations: to Boolean data [11], ordinal data [26, 4, 1], mixed data [42, 38], data from an exponential family distribution [18], and high rank matrices [16, 34, 13, 14]. However, none of these methods can quantify the uncertainty of the resulting imputations.

Multiple imputation (MI) requires repeating an imputation procedure many times to assess empirical uncertainty, often through bootstrap sampling [23, 2] or Bayesian posterior sampling [5] including probabilistic matrix factorization [32, 40]. The repeating procedure often leads to very expensive computation, especially for large datasets. While variational inference can accelerate the process in some cases [27], it may produce inaccurate results due to using overly simple approximation. Moreover, *proper* multiple imputation generally relies on strong distributional assumptions [32, 40]. In contrast, our quantified uncertainty estimates are useful for a much broader family of distributions and can be computed as fast as a single imputation. In addition, few MI papers explicitly explore the issue of *calibration*: does MI sample variance predict imputation accuracy? We find that the answer is usually no. In contrast, our uncertainty metric is clearly correlated with imputation accuracy.

Some interesting new approaches [7, 8] discuss constructing *honest* confidence regions, which depends on some (possibly huge) hidden constants. However, these unknown hidden constants prevent its use in practice. In contrast, our constructed confidence intervals are explicit.

Researchers from a Bayesian tradition have also studied the LRGC model with missing data [33, 10]. However, the associated MCMC algorithms are expensive and do not scale to large-scale data.

## 2   Notation and background

**Notation**   Let $[p] = \{1, \ldots, p\}$ for $p \in \mathbb{N}^+$. For a vector $\mathbf{x} \in \mathbb{R}^p$ and a matrix $\mathbf{W} \in \mathbb{R}^{p \times k} (p > k)$, with a set $I \subset [p]$, we denote the subvector of $\mathbf{x}$ with entries in $I$ as $\mathbf{x}_I$, and the submatrix of $\mathbf{W}$ with rows in $I$ as $\mathbf{W}_I$. Let $\mathbf{X} \in \mathbb{R}^{n \times p}$ be a matrix whose rows correspond to observations and columns to variables. We refer to the $i$-th row, $j$-th column, and $(i, j)$-th entry as $\mathbf{x}^i, \mathbf{X}_j$ and $x_j^i$, respectively. Denote the vector $\ell_2$ norm as $|| \cdot ||_2$ and the matrix Frobenius norm as $|| \cdot ||_F$.

**Gaussian copula** The Gaussian copula models any random vector supported on an (entrywise) ordered set [29, 21, 12, 15, 45] including continuous (real-valued), ordinal, and binary observations, by transforming a latent Gaussian vector. We follow the definition introduced in [45].

**Definition 1.** We say a random vector $\mathbf{x} = (x_1, \ldots, x_p) \in \mathbb{R}^p$ follows the Gaussian copula $\mathbf{x} \sim \mathrm{GC}(\boldsymbol{\Sigma}, \mathbf{g})$ with parameters $\boldsymbol{\Sigma}$ and $\mathbf{g}$ if there exists a correlation matrix $\boldsymbol{\Sigma}$ and elementwise monotone function $\mathbf{g} : \mathbb{R}^p \to \mathbb{R}^p$ such that $\mathbf{g}(\mathbf{z}) := (g_1(z_1), \ldots, g_p(z_p)) = \mathbf{x}$ for $\mathbf{z} \sim \mathcal{N}_p(\mathbf{0}, \boldsymbol{\Sigma})$.

Without loss of generality, we only consider increasing $g_j$ in this paper. By matching the marginals of $x_j$ and $g_j(z_j)$, one can show $g_j(z) = F_j(\Phi^{-1}(z))$, where $F_j$ is the cumulative distribution function (CDF) of $x_j$ for $j \in [p]$, and $\Phi$ is the standard Gaussian CDF. Thus $x_j$ has continuous distribution if and only if $g_j$ is strictly monotone. If $x_j$ is ordinal with $m$ levels, then $g_j$ is a step function with cut point set $\mathbf{S} : 1 + \sum_{s \in \mathbf{S}} \mathbb{1}(z > s)$ [45]. We focus on the case when $x_j$ for $j \in [p]$ are all continuous or all ordinal in this paper. The extension to continuous and ordinal mixed data is natural as in [45].

The Gaussian copula model balances structure and flexibility by separately modeling the marginals $\mathbf{g}$ and the correlations $\boldsymbol{\Sigma}$ in the data: assuming normality of the latent $\mathbf{z}$ allows for constructing confidence intervals, while the flexibility of nonparametric $\mathbf{g}$ enables highly accurate fit to the data. Define a set-valued inverse $g_j^{-1}(x_j) := \{z_j : g_j(z_j) = x_j\}$ and $\mathbf{g}_{\mathcal{O}}^{-1}(\mathbf{x}_{\mathcal{O}}) := \prod_{j \in \mathcal{O}} g_j^{-1}(x_j)$. The inverse set $g_j^{-1}(x_j)$ is a point, an interval and $\mathbb{R}$ for continuous, ordinal and missing $x_j$, respectively. The correlation matrix $\boldsymbol{\Sigma}$ is estimated based on the observed information $\mathbf{z}_{\mathcal{O}} \in \mathbf{g}_{\mathcal{O}}^{-1}(\mathbf{x}_{\mathcal{O}})$ [45].

Interestingly, the Gaussian copula imputations do not significantly overfit even with $O(p^2)$ parameters in $\boldsymbol{\Sigma}$ [45], however, the fitting algorithm scales cubically in $p$. Without introducing low rank structures to $\boldsymbol{\Sigma}$, the Gaussian copula model is limited to skinny datasets with $p$ up to a couple of hundreds.

# 3 Low rank Gaussian copula model

We introduce our model, the model-based imputation and its quantified uncertainty in Section 3.1, the estimation algorithm in Section 3.2, and analyze the MSE of our imputation estimator in Section 3.3.

## 3.1 Model specification and the associated imputation

We propose a low rank Gaussian copula model that integrates the flexible marginals of the Gaussian copula model with the low rank structure of the PPCA model [41]. To define the model, first consider a $p$-dimensional Gaussian vector $\mathbf{z} \sim \mathrm{PPCA}(\mathbf{W}, \sigma^2)$ generated from the PPCA model:

$$\mathbf{z} = \mathbf{W}\mathbf{t} + \boldsymbol{\epsilon}, \text{ where } \mathbf{t} \sim \mathcal{N}_k(\mathbf{0}, \mathbf{I}_k), \boldsymbol{\epsilon} \sim \mathcal{N}_p(\mathbf{0}, \sigma^2 \mathbf{I}_p), \mathbf{t} \text{ and } \boldsymbol{\epsilon} \text{ are independent}, \quad (1)$$

where $\mathbf{W} = [\mathbf{w}_1, \ldots, \mathbf{w}_p]^\mathsf{T} \in \mathbb{R}^{p \times k}$ with $p > k$. We say $\mathbf{x}$ follows the *low rank Gaussian copula (LRGC) model* if $\mathbf{x} \sim \mathrm{GC}(\mathbf{W}\mathbf{W}^\mathsf{T} + \sigma^2 \mathbf{I}_p, \mathbf{g})$ and $\mathbf{g}(\mathbf{z}) = \mathbf{x}$ for $\mathbf{z} \sim \mathrm{PPCA}(\mathbf{W}, \sigma^2)$.

To ensure that $\mathbf{x}$ follows the Gaussian copula, $\mathbf{z}$ must have zero mean and unit variance in all dimensions. Hence we require the covariance $\mathbf{W}\mathbf{W}^\mathsf{T} + \sigma^2 \mathbf{I}_p$ to have unit diagonal: $||\mathbf{w}_j||_2^2 + \sigma^2 = 1$ for $j \in [p]$. We summarize the LRGC model in the following definition.

**Definition 2.** We say a random vector $\mathbf{x} \in \mathbb{R}^p$ follows the low rank Gaussian copula $\mathbf{x} \sim \mathrm{LRGC}(\mathbf{W}, \sigma^2, \mathbf{g})$ with parameters $\mathbf{W} \in \mathbb{R}^{p \times k}(p > k), \sigma^2$ and $\mathbf{g}$ if (1) $\mathbf{g}$ is an elementwise monotonic function; (2) $\mathbf{W}\mathbf{W}^\mathsf{T} + \sigma^2 \mathbf{I}_p$ has unit diagonal; (3) $\mathbf{g}(\mathbf{z}) = \mathbf{x}$ for $\mathbf{z} \sim \mathrm{PPCA}(\mathbf{W}, \sigma^2)$.

To see the generality of the LRGC model, suppose $\mathbf{X}$ has iid rows $\mathbf{x}^i \sim \mathrm{LRGC}(\mathbf{W}, \sigma^2, \mathbf{g})$. Then

$$\mathbf{X} = \mathbf{g}(\mathbf{Z}) = \mathbf{g}(\mathbf{T}\mathbf{W}^\mathsf{T} + \mathbf{E}) := [g_1(\mathbf{Z}_1), \ldots, g_p(\mathbf{Z}_p)] = [g_1(\mathbf{T}\mathbf{w}_1 + \mathbf{E}_1), \ldots, g_p(\mathbf{T}\mathbf{w}_p + \mathbf{E}_p)] \quad (2)$$

where $\mathbf{Z}, \mathbf{T}, \mathbf{E}$ have rows $\mathbf{z}^i, \mathbf{t}^i, \boldsymbol{\epsilon}^i$, respectively, satisfying $\mathbf{z}^i = \mathbf{W}\mathbf{t}^i + \boldsymbol{\epsilon}^i$ and $\mathbf{g}(\mathbf{z}^i) = \mathbf{x}^i$ for $i \in [n]$. While the latent normal matrix $\mathbf{Z}$ has low rank plus noise structure, the observation matrix $\mathbf{X}$ can have high rank or ordinal entries with an appropriate choice of the marginals $\mathbf{g}$. When all marginals of $\mathbf{g}$ are linear functions in $\mathbb{R}$, the LRGC model reduces to the PPCA model.

Our method differs from LRMC and MI in that we treat one factor $\mathbf{W}$ as model parameters, but the other factor $\mathbf{T}$ as unseen random samples. With estimated $\mathbf{W}$, we analytically integrate over all $\mathbf{T}$ to obtain the imputation and quantify uncertainty. In contrast, LRMC and its generalization aim to estimate both factors $\mathbf{W}$ and $\mathbf{T}$ as model parameters, which make it hard to quantify uncertainty; MI treats both factors $\mathbf{W}$ and $\mathbf{T}$ as unseen random samples, which make the computation, such as the posterior distribution, intractable and requires expensive sampling on large datasets.

**Imputation** Suppose we have observed a few entries $\mathbf{x}_{\mathcal{O}}$ of a vector $\mathbf{x} \sim \text{LRGC}(\mathbf{W}, \sigma^2, \mathbf{g})$ with known $\mathbf{W}$, $\sigma$ and $\mathbf{g}$. We impute the missing entries $\mathbf{x}_{\mathcal{M}}$ and quantify the imputation uncertainty. We do not need to know the missing mechanism for imputation, so we defer a discussion to Section 3.2.

To impute $\mathbf{x}_{\mathcal{M}}$, we need its conditional distribution given observation $\mathbf{x}_{\mathcal{O}}$. Since $\mathbf{x}_{\mathcal{M}} = \mathbf{g}_{\mathcal{M}}(\mathbf{z}_{\mathcal{M}})$, we analyze the conditional distribution of $\mathbf{z}_{\mathcal{M}}$, the latent Gaussian vector at $\mathcal{M}$. From $\mathbf{x}_{\mathcal{O}}$, we can be sure that the latent Gaussian vector $\mathbf{z}_{\mathcal{O}} \in \mathbf{g}_{\mathcal{O}}^{-1}(\mathbf{x}_{\mathcal{O}})$ lies in a known set based on the observed entries. When all observed entries are continuous (i.e., $\mathbf{g}_{\mathcal{O}}$ is strictly monotone), the set $\mathbf{g}_{\mathcal{O}}^{-1}(\mathbf{x}_{\mathcal{O}})$ is a singleton, so $\mathbf{z}_{\mathcal{O}}$ is uniquely identifiable and the conditional distribution $\mathbf{z}_{\mathcal{M}}|\mathbf{z}_{\mathcal{O}}$ is Gaussian. However, for ordinal observations, $\mathbf{g}_{\mathcal{O}}^{-1}(\mathbf{x}_{\mathcal{O}})$ is a Cartesian product of intervals. The density of $\mathbf{z}_{\mathcal{M}}|\mathbf{x}_{\mathcal{O}}$ involves integrating $\mathbf{z}_{\mathcal{M}}|\mathbf{z}_{\mathcal{O}}$ over $\mathbf{z}_{\mathcal{O}} \in \mathbf{g}_{\mathcal{O}}^{-1}(\mathbf{x}_{\mathcal{O}})$, which is intractable for $|\mathcal{O}| > 1$. Fortunately, we can still estimate the mean and covariance of $\mathbf{z}_{\mathcal{M}}|\mathbf{x}_{\mathcal{O}}$, as stated in Lemma 1, upon which we construct imputation and quantify uncertainty. All proofs appear in the supplement.

**Lemma 1.** Suppose $\mathbf{x} \sim \text{LRGC}(\mathbf{W}, \sigma^2, \mathbf{g})$ with observations $\mathbf{x}_{\mathcal{O}}$ and missing entries $\mathbf{x}_{\mathcal{M}}$. Then for the latent normal vector $\mathbf{z}$ satisfying $\mathbf{g}(\mathbf{z}) = \mathbf{x}$, with corresponding latent subvectors $\mathbf{z}_{\mathcal{O}}$ and $\mathbf{z}_{\mathcal{M}}$,

$$\text{E}[\mathbf{z}_{\mathcal{M}}|\mathbf{x}_{\mathcal{O}}] = \mathbf{W}_{\mathcal{M}}\mathbf{M}_{\mathcal{O}}^{-1}\mathbf{W}_{\mathcal{O}}^{\mathsf{T}}\text{E}[\mathbf{z}_{\mathcal{O}}|\mathbf{x}_{\mathcal{O}}], \text{ where } \mathbf{M}_{\mathcal{O}} = \sigma^2\mathbf{I}_k + \mathbf{W}_{\mathcal{O}}^{\mathsf{T}}\mathbf{W}_{\mathcal{O}} \qquad (3)$$

$$\text{Cov}[\mathbf{z}_{\mathcal{M}}|\mathbf{x}_{\mathcal{O}}] = \sigma^2\mathbf{I}_{|\mathcal{M}|} + \sigma^2\mathbf{W}_{\mathcal{M}}\mathbf{M}_{\mathcal{O}}^{-1}\mathbf{W}_{\mathcal{M}}^{\mathsf{T}} + \mathbf{W}_{\mathcal{M}}\mathbf{M}_{\mathcal{O}}^{-1}\mathbf{W}_{\mathcal{O}}^{\mathsf{T}}\text{Cov}[\mathbf{z}_{\mathcal{O}}|\mathbf{x}_{\mathcal{O}}]\mathbf{W}_{\mathcal{O}}\mathbf{M}_{\mathcal{O}}^{-1}\mathbf{W}_{\mathcal{M}}^{\mathsf{T}} \quad (4)$$

For continuous $\mathbf{x}_{\mathcal{O}}$, the latent $\mathbf{z}_{\mathcal{O}}$ is identifiable: $\text{E}[\mathbf{z}_{\mathcal{O}}|\mathbf{x}_{\mathcal{O}}] = \mathbf{g}_{\mathcal{O}}^{-1}(\mathbf{x}_{\mathcal{O}})$ and $\text{Cov}[\mathbf{z}_{\mathcal{O}}|\mathbf{x}_{\mathcal{O}}] = \mathbf{0}$. For ordinal $\mathbf{x}_{\mathcal{O}}$, $\text{E}[\mathbf{z}_{\mathcal{O}}|\mathbf{x}_{\mathcal{O}}]$ and $\text{Cov}[\mathbf{z}_{\mathcal{O}}|\mathbf{x}_{\mathcal{O}}]$ are the mean and covariance of a truncated normal vector, determined by $\mathbf{W}_{\mathcal{O}}, \sigma^2$ and $\mathbf{g}_{\mathcal{O}}^{-1}(\mathbf{x}_{\mathcal{O}})$. We discuss the computation strategies in Section 3.2.

It is natural to impute $\mathbf{x}_{\mathcal{M}}$ by mapping the conditional mean of $\mathbf{z}_{\mathcal{M}}$ through the marginals $\mathbf{g}_{\mathcal{M}}$.

**Definition 3** (LRGC Imputation). Suppose $\mathbf{x} \sim \text{LRGC}(\mathbf{W}, \sigma^2, \mathbf{g})$ with observations $\mathbf{x}_{\mathcal{O}}$ and missing entries $\mathbf{x}_{\mathcal{M}}$. We impute the missing entries as $\hat{\mathbf{x}}_{\mathcal{M}} = \mathbf{g}_{\mathcal{M}}(\text{E}[\mathbf{z}_{\mathcal{M}}|\mathbf{x}_{\mathcal{O}}])$ with $\text{E}[\mathbf{z}_{\mathcal{M}}|\mathbf{x}_{\mathcal{O}}]$ in Lemma 1.

**Imputation uncertainty quantification** Can we quantify the uncertainty in these imputations? Different from LRMC model which assumes a deterministic true value for missing locations, $\mathbf{x}_{\mathcal{M}}$ (as well as $\mathbf{z}_{\mathcal{M}}$) is random under the LRGC model. Consequently, the error $\hat{\mathbf{x}}_{\mathcal{M}} - \mathbf{x}_{\mathcal{M}}$ is random and hence uncertain even with deterministic imputation $\hat{\mathbf{x}}_{\mathcal{M}}$. The uncertainty depends on the concentration of $\mathbf{z}_{\mathcal{M}}$ around its mean $\text{E}[\mathbf{z}_{\mathcal{M}}|\mathbf{x}_{\mathcal{O}}]$ and on the marginals $\mathbf{g}_{\mathcal{M}}$. If $\mathbf{g}_{\mathcal{M}}$ is constant or nearly constant over the likely values of $\mathbf{z}_{\mathcal{M}}$, then with high probability the imputation is accurate. Otherwise, the current observations cannot predict the missing entry well and we should not trust the imputation. Using this intuition, we may formally quantify the uncertainty in the imputations. For continuous data, we construct confidence intervals using the normality of $\mathbf{z}_{\mathcal{M}}|\mathbf{z}_{\mathcal{O}} = \mathbf{g}_{\mathcal{O}}^{-1}(\mathbf{x}_{\mathcal{O}})$.

**Theorem 1** (Uncertainty quantification for continuous data). Suppose $\mathbf{x} \sim \text{LRGC}(\mathbf{W}, \sigma^2, \mathbf{g})$ with observations $\mathbf{x}_{\mathcal{O}}$ and missing entries $\mathbf{x}_{\mathcal{M}}$ and that $\mathbf{g}$ is elementwise strictly monotone. For missing entry $x_j$, for any $\alpha \in (0, 1)$, let $z^{\star} = \Phi^{-1}(1 - \frac{\alpha}{2})$, the following holds with probability $1 - \alpha$:

$$x_j \in [g_j(\text{E}[z_j|\mathbf{x}_{\mathcal{O}}] - z^{\star}\text{Var}[z_j|\mathbf{x}_{\mathcal{O}}]), g_j(\text{E}[z_j|\mathbf{x}_{\mathcal{O}}] + z^{\star}\text{Var}[z_j|\mathbf{x}_{\mathcal{O}}])] =: [x_j^-(\alpha), x_j^+(\alpha)] \qquad (5)$$

where $\text{E}[z_j|\mathbf{x}_{\mathcal{O}}], \text{Var}[z_j|\mathbf{x}_{\mathcal{O}}]$ are given in Lemma 1 with $\mathcal{M}$ replaced by $j$, for $j \in \mathcal{M}$.

For ordinal data, we lower bound the probability of correct prediction $x_j = \hat{x}_j$ using a sufficient condition that $z_j$ is sufficiently close to its mean $\text{E}[z_j|\mathbf{x}_{\mathcal{O}}]$. General results in bounding $\Pr(|\hat{x}_j - x_j| \leq d)$ for any $d \in \mathbb{Z}$ appear in the supplement. Note a step function $g_j(z)$ with cut points set admits the form $\mathbf{S}$: $g_j(z) = 1 + \sum_{s \in \mathbf{S}} \mathbb{1}(z > s)$.

**Theorem 2** (Uncertainty quantification for ordinal data). Suppose $\mathbf{x} \sim \text{LRGC}(\mathbf{W}, \sigma^2, \mathbf{g})$ with observations $\mathbf{x}_{\mathcal{O}}$ and missing entries $\mathbf{x}_{\mathcal{M}}$ and that the marginal $g_j$ is a step function with cut points $\mathbf{S}_j$, for $j \in [p]$. For missing entry $x_j$ and its imputation $\hat{x}_j = g_j(\text{E}[z_j|\mathbf{x}_{\mathcal{O}}])$,

$$\Pr(\hat{x}_j = x_j) \geq 1 - \text{Var}[z_j|\mathbf{x}_{\mathcal{O}}]/d_j^2 \quad \text{where} \quad d_j = \min_{s \in \mathbf{S}_j} |s - \text{E}[z_j|\mathbf{x}_{\mathcal{O}}]|, \qquad (6)$$

and $\text{E}[z_j|\mathbf{x}_{\mathcal{O}}], \text{Var}[z_j|\mathbf{x}_{\mathcal{O}}]$ are given in Lemma 1 with $\mathcal{M}$ replaced by $j$, for $j \in \mathcal{M}$.

To predict the imputation accuracy using the quantified uncertainty, we develop a measure we call *reliability*. Entries with higher reliability are expected to have smaller imputation error. We first motivate our definition of reliability. For ordinal data, the reliability of an entry lower bounds the

probability of correct prediction. For continuous data, our measure of reliability is designed so that reliable imputations have low normalized root mean squared error (NRMSE) under a certain confidence level $\alpha$. NRMSE is defined as $||P_{\Omega^c}(\mathbf{X} - \hat{\mathbf{X}})||_F / ||P_{\Omega^c}(\mathbf{X})||_F$ for matrix $\mathbf{X}$ observed on $\Omega$ and its imputed matrix $\hat{\mathbf{X}}$. Here $P_\Omega$ is projection onto the set $\Omega$: it sets entries not in $\Omega$ to 0.

Our definition of reliability uses Theorems 1-2 to ensure that reliable imputations have low error.

**Definition 4** (LRGC Imputation Reliability). Suppose $\mathbf{X}$ has iid rows $\mathbf{x}^i \sim \text{LRGC}(\mathbf{W}, \sigma^2, \mathbf{g})$ and is observed on $\Omega \subset [n] \times [p]$. Complete $\mathbf{X}$ to $\hat{\mathbf{X}}$ row-wise using Definition 3. For each missing entry $(i, j) \in \Omega^c$, define the *reliability* of the imputation $\hat{x}_j^i$ as

- (if $\mathbf{X}$ is an ordinal matrix) the lower bound provided in Eq. (6);
- (if $\mathbf{X}$ is a continuous matrix) $||P_{\Omega^c \backslash (i,j)}(D_\alpha)||_F / ||P_{\Omega^c \backslash (i,j)}(\hat{\mathbf{X}})||_F$, where the $(i', j')$-th entry of matrix $D_\alpha$ is the length of the confidence interval $\hat{x}_{j'}^{i',+}(\alpha) - \hat{x}_{j'}^{i',-}(\alpha)$ defined in Eq. (5).

For continuous data, the interpretation is that if the error after removing $(i_1, j_1)$ is larger than that after removing $(i_2, j_2)$, then the imputation on $(i_1, j_1)$ is more reliable than that on $(i_2, j_2)$. If continuous entries in different columns are measured on very different scales, one can also modify the definition to compute reliability column-wise.

Our experiments show this reliability measure positively correlates with imputation accuracy as measured by mean absolute error (MAE) for ordinal data and NRMSE for continuous data. We also find for continuous data, the correlation is insensitive to $\alpha$ in a reasonable range; $\alpha = .05$ works well.

## 3.2 Fitting the low rank Gaussian copula

Suppose $\mathbf{X} \in \mathbb{R}^{n \times p}$ observed on $\Omega$ has iid rows $\mathbf{x}^i \sim \text{LRGC}(\mathbf{W}, \sigma^2, \mathbf{g})$ and $\mathbf{x}^i$ has observations $\mathbf{x}_{\mathcal{O}_i}^i$ and missing entries $\mathbf{x}_{\mathcal{M}_i}^i$. As in prior work [29, 19, 45], we estimate $\mathbf{g}$ by matching it to the empirical distribution of observations. To estimate $(\mathbf{W}, \sigma^2)$, we propose an EM algorithm that scales linearly in $n$ and $p$, using ingredients from [45, 19] for the E-step, and from [22] for the M-step. Due to space limit, we present essentials here and summarize in Algorithm 1. Details appear in the supplement.

**Estimate the marginals** We need the marginals $g_j$ for imputation and their inverse $g_j^{-1}$ to estimate the covariance. Noticing $g_j$ maps standard normal data to $\mathbf{X}_j$ which has CDF $F_j$, thus we may estimate these functions by the empirical distribution of observations in $\mathbf{X}_j$. Concretely, recall that $g_j = F_j^{-1} \circ \Phi$ and $g_j^{-1} = \Phi^{-1} \circ F_j$. We estimate $g_j^{-1}$ by replacing $F_j$ with the scaled empirical CDF $\frac{n}{n+1}\hat{F}_j$, where the scaling $\frac{n}{n+1}$ is chosen to ensure finite output of $g_j^{-1}$. Similarly, we estimate $g_j$ using the empirical quantiles of observations in $\mathbf{X}_j$. To ensure that the empirical CDF consistently estimates the true CDF, we assume the missing completely at random mechanism (MCAR). It is possible to relax the it to missing at random (MAR) or even missing not at random by modeling either $F_j$ or the missing mechanism. We leave this important work for the future.

**EM algorithm for $\mathbf{W}$ and $\sigma^2$** Ideally, we would compute the maximum likelihood estimates (MLE) for the copula parameters $(\mathbf{W}, \sigma^2)$ (under the likelihood in Eq. (7)), which are consistent under the MAR mechanism [28, Chapter 6.2] as $n \to \infty$. However, the likelihood involves a Gaussian integral that is hard to optimize. Instead, we estimate the MLE using an approximate EM algorithm.

The likelihood of $(\mathbf{W}, \sigma^2)$ given observation $\mathbf{x}_{\mathcal{O}_i}^i$ is the integral over the latent Gaussian vector $\mathbf{z}_{\mathcal{O}_i}^i$ that maps to $\mathbf{x}_{\mathcal{O}_i}^i$ under the marginal $\mathbf{g}_{\mathcal{O}_i}$. Hence the observed log likelihood we seek to maximize is:

$$\ell_{\text{obs}}(\mathbf{W}, \sigma^2; \{\mathbf{x}_{\mathcal{O}_i}^i\}_{i=1}^n) = \sum_{i=1}^n \log \int_{\mathbf{z}_{\mathcal{O}_i}^i \in \mathbf{g}_{\mathcal{O}_i}^{-1}(\mathbf{x}_{\mathcal{O}_i}^i)} \phi(\mathbf{z}_{\mathcal{O}_i}^i; \mathbf{0}, \mathbf{W}_{\mathcal{O}_i}\mathbf{W}_{\mathcal{O}_i}^{\mathsf{T}} + \sigma^2 \mathbf{I}_{|\mathcal{O}_i|}) d\mathbf{z}_{\mathcal{O}_i}^i, \quad (7)$$

where $\phi(\cdot; \boldsymbol{\mu}, \boldsymbol{\Sigma})$ denotes the Gaussian vector density with mean $\boldsymbol{\mu}$ and covariance $\boldsymbol{\Sigma}$. Recall the decomposition $\mathbf{Z} = \mathbf{T}\mathbf{W}^{\mathsf{T}} + \mathbf{E}$ as in Eq. (2). If $\mathbf{z}_{\mathcal{O}_i}^i$ and $\mathbf{t}^i$ are known, the joint likelihood is simple:

$$\ell(\mathbf{W}, \sigma^2; \{\mathbf{x}_{\mathcal{O}_i}^i, \mathbf{z}_{\mathcal{O}_i}^i, \mathbf{t}^i\}_{i=1}^n) = \sum_{i=1}^n \log \left[ \phi(\mathbf{z}_{\mathcal{O}_i}^i; \mathbf{W}_{\mathcal{O}_i}\mathbf{t}^i, \sigma^2 \mathbf{I}_p) \, \phi(\mathbf{t}^i; \mathbf{0}, \mathbf{I}_k) \mathbb{1}_{\mathbf{g}_{\mathcal{O}_i}^{-1}(\mathbf{x}_{\mathcal{O}_i}^i)}(\mathbf{z}_{\mathcal{O}_i}^i) \right]. \quad (8)$$

Here define $\mathbb{1}_A(x) = 1$ when $x \in A$ and $0$ otherwise. The maximizers $(\hat{\mathbf{W}}, \hat{\sigma})$ of Eq. (8) are $\hat{\mathbf{W}} = \operatorname{argmin}_{\mathbf{W}} ||P_\Omega(\mathbf{Z} - \mathbf{TW}^\intercal)||_F^2$ and $\hat{\sigma}^2 = ||P_\Omega(\mathbf{Z} - \mathbf{T}\hat{\mathbf{W}}^\intercal)||_F^2 / |\Omega|$. Moreover, the problem is separable over the rows of $\hat{\mathbf{W}}$: to solve for the $j$-th row $\hat{\mathbf{w}}_j^\intercal$, we use only $\mathbf{z}_{\mathcal{O}_i}^i, \mathbf{t}^i$ for $i \in \Omega_j = \{i : (i,j) \in \Omega\}$. Our EM algorithm treats the unknown $\mathbf{z}_{\mathcal{O}_i}^i, \mathbf{t}^i$ as latent variables and $\mathbf{x}_{\mathcal{O}_i}^i$ as the observed variable. Given an estimate $(\tilde{\mathbf{W}}, \tilde{\sigma}^2)$, the E-step computes the expectation $\mathbb{E}[||P_\Omega(\mathbf{Z} - \mathbf{TW}^\intercal)||_F^2]$ with respect to $\mathbf{z}_{\mathcal{O}_i}^i$ and $\mathbf{t}^i$ conditional on $\mathbf{x}_{\mathcal{O}_i}^i$. Throughout the section, we use $\mathbb{E}$ to denote this conditional expectation. The M-step is similar to when $\mathbf{z}_{\mathcal{O}_i}^i$ and $\mathbf{t}^i$ are known.

**E-step** Calculate the expected likelihood $Q(\mathbf{W}, \sigma^2; \tilde{\mathbf{W}}, \tilde{\sigma}^2) = \mathbb{E}[\ell(\mathbf{W}, \sigma^2; \{\mathbf{x}_{\mathcal{O}_i}^i, \mathbf{z}_{\mathcal{O}_i}^i, \mathbf{t}^i\}_{i=1}^n)]$. It suffices to compute $\mathbb{E}[(\mathbf{z}_{\mathcal{O}_i}^i)^\intercal \mathbf{z}_{\mathcal{O}_i}^i], \mathbb{E}[\mathbf{t}^i (\mathbf{z}_{\mathcal{O}_i}^i)^\intercal]$ and $\mathbb{E}[\mathbf{t}^i (\mathbf{t}^i)^\intercal]$ as detailed below.

**M-step** Let $\mathbf{e}_j \in \mathbb{R}^p$ be the $j$th standard basis vector. The maximizers of $Q(\mathbf{W}, \sigma^2; \tilde{\mathbf{W}}, \tilde{\sigma}^2)$ are

$$\hat{\mathbf{w}}_j^\intercal = \left( \mathbf{e}_j^\intercal \sum_{i \in \Omega_j} \mathbb{E}[\mathbf{z}_{\mathcal{O}_i}^i (\mathbf{t}^i)^\intercal] \right) \left( \sum_{i \in \Omega_j} \mathbb{E}[\mathbf{t}^i (\mathbf{t}^i)^\intercal] \right)^{-1}, \quad \hat{\sigma}^2 = \frac{\sum_{i=1}^n \mathbb{E}\left[ ||\mathbf{z}_{\mathcal{O}_i}^i - \hat{\mathbf{W}}_{\mathcal{O}_i} \mathbf{t}^i)||_2^2 \right]}{\sum_{i=1}^n |\mathcal{O}_i|}. \quad (9)$$

The maximizer $(\hat{\mathbf{W}}, \hat{\sigma}^2)$ increase the observed likelihood in Eq. (7) compared to the initial estimate $(\tilde{\mathbf{W}}, \tilde{\sigma}^2)$ [31, Chapter 3]. To satisfy the unit diagonal constraints $||\mathbf{w}_j||_2^2 + \sigma^2 = 1$, we approximate the constrained maximizer by scaling the unconstrained maximizer shown in Eq. (10) as in [19, 45]:

$$\hat{\sigma}^2 \leftarrow \hat{\sigma}_{\text{new}}^2 = \frac{1}{p} \sum_{j=1}^p \frac{\hat{\sigma}^2}{||\hat{\mathbf{w}}_j||_2^2 + \hat{\sigma}^2}, \quad \hat{\mathbf{w}}_j \leftarrow \frac{\hat{\mathbf{w}}_j}{||\hat{\mathbf{w}}_j||_2} \cdot \sqrt{1 - \hat{\sigma}_{\text{new}}^2}. \quad (10)$$

We find this approximation works well in practice.

**Computation** We can express all expectations in the E-step using $\mathrm{E}[\mathbf{z}_{\mathcal{O}_i}^i | \mathbf{x}_{\mathcal{O}_i}^i]$ and $\mathrm{Cov}[\mathbf{z}_{\mathcal{O}_i}^i | \mathbf{x}_{\mathcal{O}_i}^i]$:

$$\mathbb{E}[\mathbf{t}^i] = \mathbf{M}_{\mathcal{O}_i}^{-1} \mathbf{W}_{\mathcal{O}_i}^\intercal \mathrm{E}[\mathbf{z}_{\mathcal{O}_i}^i | \mathbf{x}_{\mathcal{O}_i}^i], \quad \text{where } \mathbf{M}_{\mathcal{O}_i} = \sigma^2 \mathbf{I}_k + \mathbf{W}_{\mathcal{O}_i}^\intercal \mathbf{W}_{\mathcal{O}_i}. \quad (11)$$

$$\mathbb{E}[\mathbf{t}^i (\mathbf{z}_{\mathcal{O}_i}^i)^\intercal] = \mathbb{E}[\mathbf{t}^i] \mathrm{E}[\mathbf{z}_{\mathcal{O}_i}^i | \mathbf{x}_{\mathcal{O}_i}^i]^\intercal + \mathbf{M}_{\mathcal{O}_i}^{-1} \mathbf{W}_{\mathcal{O}_i}^\intercal \mathrm{Cov}[\mathbf{z}_{\mathcal{O}_i}^i | \mathbf{x}_{\mathcal{O}_i}^i]. \quad (12)$$

$$\mathbb{E}[\mathbf{t}^i (\mathbf{t}^i)^\intercal] = \sigma^2 \mathbf{M}_{\mathcal{O}_i}^{-1} + \mathbb{E}[\mathbf{t}^i] \mathbb{E}[\mathbf{t}^i]^\intercal + \mathbf{M}_{\mathcal{O}_i}^{-1} \mathbf{W}_{\mathcal{O}_i}^\intercal \mathrm{Cov}[\mathbf{z}_{\mathcal{O}_i}^i | \mathbf{x}_{\mathcal{O}_i}^i] \mathbf{W}_{\mathcal{O}_i} \mathbf{M}_{\mathcal{O}_i}^{-1}. \quad (13)$$

For continuous data, recall $\mathrm{E}[\mathbf{z}_{\mathcal{O}_i}^i | \mathbf{x}_{\mathcal{O}_i}^i] = \mathbf{g}_{\mathcal{O}_i}^{-1}(\mathbf{x}_{\mathcal{O}_i}^i)$ and $\mathrm{Cov}[\mathbf{z}_{\mathcal{O}_i}^i | \mathbf{x}_{\mathcal{O}_i}^i] = \mathbf{0}$. For ordinal data, these quantities are the mean and covariance of a truncated normal vector, for each row $i$ separately at each EM iteration. Direct computation [3] or sampling methods [35] are expensive for large $n, p$. We follow instead the fast iterative method in [19, 45]. The intuition is that for each $j \in \mathcal{O}_i$, conditional on known $\mathbf{z}_{\mathcal{O}_i \setminus \{j\}}$ and the constraint $z_j^i \in g_j^{-1}(x_j^i)$, $z_j^i$ is univariate truncated normal with closed-form mean and variance. Thus one can iteratively update the marginal mean and variance of $\mathbf{z}_{\mathcal{O}_i}^i | \mathbf{x}_{\mathcal{O}_i}^i$. The rigorous formulation iteratively solves a nonlinear system satisfied by $\mathrm{E}[\mathbf{z}_{\mathcal{O}_i}^i | \mathbf{x}_{\mathcal{O}_i}^i]$ and by diagonals of $\mathrm{Cov}[\mathbf{z}_{\mathcal{O}_i}^i | \mathbf{x}_{\mathcal{O}_i}^i]$ respectively, detailed in the supplement. Estimating the off-diagonals of $\mathrm{Cov}[\mathbf{z}_{\mathcal{O}_i}^i | \mathbf{x}_{\mathcal{O}_i}^i]$ efficiently for large $n, p$ is still an open problem: prior work approximates all off-diagonals as 0 [19, 45]. It is showed this diagonal approximation [45] yields more accurate and much faster parameters estimate than an alternative Bayesian algorithm without the approximation [21]. The approximation also reduces the computation for Eq. (13) from $O(|\mathcal{O}|^2 k)$ to $O(|\mathcal{O}| k^2)$.

The computational complexity for each iteration is $O(|\Omega| k^2 + n k^3 + p k^3)$, upper bounded by $O(npk^2)$. We find the method usually converges in fewer than 50 iterations across our experiments. See the Movielens 1M experiment in Section 4 for a run time comparison with state-of-the-are methods.

## 3.3 Imputation error bound

The imputation error consists of two parts: (1) the random variation of the error under the true LRGC model; (2) the estimation error of the LRGC model. Analyzing the estimation error (2) is challenging for output from EM algorithm; moreover, in our experiments we find that the imputation error can be attributed predominantly to (1), detailed in the supplement. Hence we leave (2) to future work. To analyze the random variation of the error under the true LRCG model, we examine the MSE of $\hat{\mathbf{x}}$ for a random row $\mathbf{x} \sim \mathrm{LRGC}(\mathbf{W}, \sigma^2, \mathbf{g})$ with fixed missing locations $\mathcal{M}$: $\mathrm{MSE}(\hat{\mathbf{x}}) =$

**Algorithm 1** Imputation via low rank Gaussian copula fitting

---

**Input:** $\mathbf{X} \in \mathbb{R}^{n \times p}$ observed on $\Omega$, rank $k$, $t_{\max}$.

1: Compute the empirical CDF $\hat{F}_j$ and empirical quantile function $\hat{F}_j^{-1}$ on observed $\mathbf{X}_j$, for $j \in [p]$.
2: Estimate $\hat{g}_j = \Phi^{-1} \circ \frac{n}{n+1} \hat{F}_j$ and $\hat{g}_j^{-1} = \hat{F}_j^{-1} \circ \Phi$, for $j \in [p]$.
3: Initialize: $\mathbf{W}^{(0)}, (\sigma^2)^{(0)}$
4: **for** $t = 1, 2, \ldots, t_{\max}$ **do**
5:     E-step: compute the required conditional expectation using Eq. (11-13).
6:     M-step: update $\mathbf{W}^{(t)}, (\sigma^2)^{(t)}$ using Eq. (9-10).
7: **end for**
8: Impute $\hat{\mathbf{x}}_{\mathcal{M}_i}^i$ using Definition 3 for $i \in [n]$ with $\mathbf{g} = \hat{\mathbf{g}}, \mathbf{W} = \mathbf{W}^{(t_{\max})}, \sigma^2 = (\sigma^2)^{(t_{\max})}$.

**Output:** $\hat{\mathbf{X}}$ with imputed $\hat{x}_j^i$ at $(i, j) \in \Omega^c$ and observed $x_j^i$ at $(i, j) \in \Omega$.

---

$||\mathbf{g}_{\mathcal{M}}(\hat{\mathbf{z}}_{\mathcal{M}}) - \mathbf{g}_{\mathcal{M}}(\mathbf{z}_{\mathcal{M}})||_2^2 / |\mathcal{M}|$. For continuous $\mathbf{x}$ with strictly monotone $\mathbf{g}$, we must assume that $\mathbf{g}$ is Lipschitz to obtain a finite bound on the error. With this assumption and assuming $\mathbf{W}, \sigma^2$ fixed and known, we can use the fact that $\mathbf{z}_{\mathcal{M}} | \mathbf{x}_{\mathcal{O}}$ is normal to bound large deviations of the MSE.

**Theorem 3.** Suppose subvector $\mathbf{x}_{\mathcal{O}}$ of $\mathbf{x} \sim \text{LRGC}(\mathbf{W}, \sigma^2, \mathbf{g})$ is observed and that all marginals $\mathbf{g}$ are strictly monotone with Lipschitz constant $L$. Denote the largest and the smallest singular values of $\mathbf{W}'$ as $\lambda_1(\mathbf{W}')$ and $\lambda_k(\mathbf{W}')$. Then for any $t > 0$, the imputed values $\hat{\mathbf{x}}$ in Definition 3 satisfy

$$\Pr\left[ \text{MSE}(\hat{\mathbf{x}}) > L^2\sigma^2 \left( \sqrt{1 + \frac{1-\sigma^2}{\sigma^2 + \lambda_k^2(\mathbf{W}_{\mathcal{O}})}} + \sqrt{2\left(1 + \frac{\lambda_1^2(\mathbf{W}_{\mathcal{M}})}{\sigma^2 + \lambda_k^2(\mathbf{W}_{\mathcal{O}})}\right)\frac{t}{|\mathcal{M}|}} \right)^2 \right] \le e^{-t}.$$

Theorem 3 indicates the imputation error concentrates at $\sigma^2 + \frac{\sigma^2(1-\sigma^2)}{\sigma^2 + \lambda_k^2(\mathbf{W}_{\mathcal{O}})/\sigma^2}$ with an expansion multiplier $L^2$ due to the marginals $\mathbf{g}$. The first term $\sigma^2$ represents the fraction of variance due to noise and the second term is small when the SNR is large. We also analyze the distribution of $\lambda_k^2(\mathbf{W}_{\mathcal{O}})/\sigma^2$ under a random design to provide insight into when the error is small: in Corollary 1, the second term vanishes with increasing observed length $|\mathcal{O}|$.

**Corollary 1.** Under the conditions of Theorem 3, further assume $\mathbf{W}$ has independent sub-Gaussian rows $\mathbf{w}_j$ with zero mean and covariance $\frac{1-\sigma^2}{k}\mathbf{I}_k$ for $j \in [p]$. Suppose $c_1 k < |\mathcal{O}| < c_2|\mathcal{M}|$ for some constant $c_1 > 0$ depending on the sub-Gaussian norm of the scaled rows $\sqrt{\frac{k}{1-\sigma^2}}\mathbf{w}_j$ and some absolute constant $c_2 > 0$. Then for some constant $c_3 > 0$ depending on $c_1, c_2$,

$$\Pr\left[ \text{MSE}(\hat{\mathbf{x}}) > L^2\sigma^2 \left(1 + K_{|\mathcal{O}|}\right) \right] \le c_3/|\mathcal{O}|, \quad \text{where } K_{|\mathcal{O}|} = O\left(\sqrt{\log(|\mathcal{O}|)/|\mathcal{O}|}\right). \quad (14)$$

See the supplement for definition of a sub-Gaussian vector. Analyzing the imputation error for ordinal $\mathbf{x}$ is much harder since $\mathbf{z}_{\mathcal{M}} | \mathbf{x}_{\mathcal{O}}$ is no longer Gaussian. We leave that for future work.

## 4 Experiments and conclusion

Our experiments evaluate the imputation accuracy of `LRGC`, whether our reliability measure (denoted as `LRGC` reliability) can predict imputation accuracy well, and the empirical coverage of our proposed confidence intervals. For the second task, we evaluate the imputation on the $m\%$ entries with highest reliability for varying $m$. We say a measure predicts imputation accuracy if the imputation error on the $m\%$ entries is smaller for smaller $m$, i.e., it positively correlates with imputation accuracy. We introduce below competitors for each task. Implementation details appear in the supplement.

For imputation comparison, we implement LRMC methods `softImpute` [30], GLRM [42] with $\ell_2, \ell_1$, bigger vs smaller (BvS, for ordinal data), hinge, and logistic loss. We also implement the high rank matrix completion method `MMC` [16], and `PPCA`, a special case of `LRGC` with Gaussian marginals. To measure the imputation error, we use NRMSE for continuous data and MAE for ordinal data.

For reliability comparison, we compare with variance based reliability: the imputation for a given missing entry is more reliable if it has smaller variance. To obtain variance estiamte, we implement the

Table 1: Imputation error (NRMSE for continuous and MAE for ordinal) reported over 20 repetitions, with rank $r$ for available methods. `GLRM` methods are trained at rank 199.

| Continuous | LRGC | PPCA | softImpute | GLRM-$\ell_2$ | MMC |
|---|---|---|---|---|---|
| Low Rank | $.347(.004), r = 10$ | $\mathbf{.338}(\mathbf{.004}), r = 10$ | $.371(.004), r = 117$ | $.364(.003)$ | $.633(.007), r = 130$ |
| High Rank | $\mathbf{.517}(\mathbf{.011}), r = 10$ | $.690(.010), r = 10$ | $.703(.005), r = 104$ | $.696(.006)$ | $.824(.011), r = 137$ |

| 1-5 ordinal | LRGC | PPCA | softImpute | GLRM-BvS | GLRM-$\ell_1$ |
|---|---|---|---|---|---|
| High SNR | $\mathbf{.358}(\mathbf{.008}), r = 5$ | $.501(.010), r = 6$ | $.582(.011), r = 83$ | $.407(.007)$ | $.689(.010)$ |
| Low SNR | $\mathbf{.788}(\mathbf{.013}), r = 5$ | $.863(.013), r = 5$ | $.951(.015), r = 38$ | $.850(.011)$ | $1.027(.020)$ |

| Binary | LRGC | PPCA | softImpute | GLRM-hinge | GLRM-logistic |
|---|---|---|---|---|---|
| High SNR | $\mathbf{.103}(\mathbf{.003}), r = 5$ | $.116(.002), r = 6$ | $.136(.003), r = 71$ | $.140(.002)$ | $.117(.002)$ |
| Low SNR | $\mathbf{.205}(\mathbf{.006}), r = 5$ | $.208(.005), r = 5$ | $.234(.007, r = 61$ | $.226(.006)$ | $.217(.005)$ |

PCA based MI method (denoted as `MI-PCA`) [23, 24], and construct MI style uncertainty quantification for general imputation algorithms: given an algorithm and incomplete $\mathbf{X}$, divide the observations into $N$ parts. Then apply the algorithm $N$ times, each time additionally masking one part of the observations. Compute the variance of the original missing entries across $N$ estimates. We use $N = 10$ in this paper. We denote such methods as `MI+Algorithm` for applied algorithm.

For confidence interval (CI) comparison, we compare with CI based on the `softImpute` imputation (denoted as `LRMC`) [9] , CI based on `PPCA` and CI based on `MI-PCA`. All constructed intervals except `LRGC` are derived assuming normality: specifically, they all assume $\mathbf{X} = \mathbf{X}^\star + \mathbf{E}$ for some low rank $\mathbf{X}^\star$ and isotropic Gaussian error $\mathbf{E}$. In particular, their CIs are always symmetric around the imputed value, while `LRGC` can yield asymmetrical CIs. See the supplement for implementation details.

**Synthetic experiments** We consider three data types from LRGC: continuous, 1-5 ordinal and binary. We generate $\mathbf{W} \in \mathbb{R}^{p \times k}, \mathbf{T} \in \mathbb{R}^{n \times k}, \mathbf{E} \in \mathbb{R}^{n \times p}$ with independent standard normal entries, then scale each row of $\mathbf{W}$ such that $||\mathbf{w}_j||_2^2 + \sigma^2 = 1$. Then generate $\mathbf{X} = \mathbf{g}(\mathbf{Z}) = \mathbf{g}(\mathbf{T}\mathbf{W}^\intercal + \sigma\mathbf{E})$ using $\mathbf{g}$ described below. Missing entries of $\mathbf{X}$ are uniformly sampled. We set $n = 500$ and $p = 200$. For continuous data, we use $g_j(z) = z$ to generate a low rank $\mathbf{X} = \mathbf{Z}$ and $g_j(z) = z^3$ to generate a high rank $\mathbf{X}$. We set $k = 10, \sigma^2 = 0.1$ and the missing ratio as $40\%$. For 1-5 ordinal data and binary data, we use step functions $g_j$ with random selected cut points. We generate one $\mathbf{X}$ with high SNR $\sigma^2 = 0.1$ and one $\mathbf{X}$ with low SNR $\sigma^2 = 0.5$. We set $k = 5$ and the missing ratio as $60\%$.

We examine the sensitivity of each method to its key tuning parameter. Both `LRGC` and `PPCA` do not overfit with large ranks. We report results using the best tuning parameter in Table 1. The complete results and implementation details appear in the supplement. All experiments are repeated 20 times.

Shown in Table 1, `LRGC` performs the best in all but one settings. The improvement is significant for high rank continuous data. For low rank continuous data, `PPCA` performs the best as expected since the model is correctly specified. The slightly larger error of `LRGC` is due to the error in estimating a nonparametric marginal $\mathbf{g}$. Notice both `LRGC` and `PPCA` admit much smaller rank as best parameter.

Shown in Figure 1, `LRGC` reliability predicts the imputation accuracy well: entries with higher reliability (smaller $m$) have higher accuracy. In contrast, entries with higher variance based reliability can have lower accuracy. Even when the variance based reliability predicts accuracy, `LRGC` reliability works better: the error over selected entries using variance based reliability is much larger than that of `LRGC` reliability when a small percentage of entries $m$ are selected. `LRGC` reliability can even find entries with error near 0 from very noisy (low SNR 1-5 ordinal and binary) data. `LRGC` reliability better predicts imputation error for easier imputation tasks (lower rank and higher SNR). Predicting NRMSE is challenging, since imputing continuous data is in general harder than imputing ordinal data. In fact, we show in the supplement that as the number of levels of the ordinal variable increases, the shape of the error vs reliability curve matches that of continuous data.

The results on confidence intervals appear in Table 2. Notice constructing `MI-PCA` intervals is much more expensive than all other methods. For low rank Gaussian data, `PPCA` confidence intervals achieve the highest coverage rates with smallest length as expected, since the model is correctly specified. `LRGC` confidence intervals have slightly smaller coverage rates due to the error in estimating a nonparametric marginal $\mathbf{g}$. For high rank data, the normality and the low rank assumption do not hold, so all other constructed confidence intervals but `LRGC` are no longer theoretically valid. Notably,

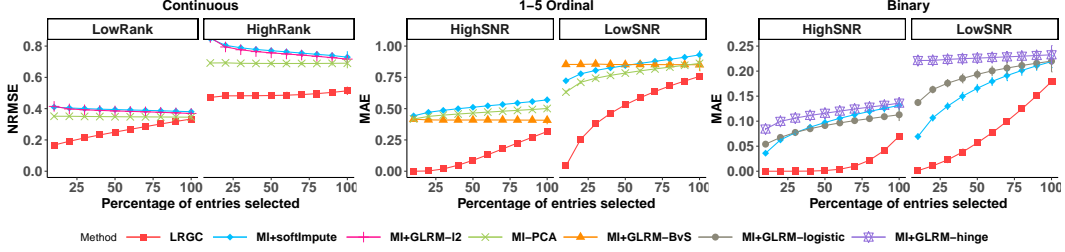

Figure 1: Imputation error on the subset of $m\%$ entries for which method's associated uncertainty metric indicates highest reliability, reported over 20 repetitions (error bars almost invisible).

Table 2: 95% Confidence intervals on synthetic continuous data over 20 repetitions.

| Low Rank Data | LRGC | PPCA | LRMC | MI-PCA |
|---|---|---|---|---|
| Empirical coverage rate | 0.927(.002) | 0.940(.001) | 0.878(.006) | 0.933(.002) |
| Interval length | 1.273(.004) | 1.264(.004) | 1.129(.015) | 1.267(.004) |
| Run time (in seconds) | 6.9(.5) | 3.4(.7) | 2.7(.4) | 189.8(15.4) |
| High Rank Data | LRGC | PPCA | LRMC | MI-PCA |
| Empirical coverage rate | 0.927(.002) | 0.943(.002) | 0.925(.004) | 0.948(.002) |
| Interval length | 3.614(.068) | 9.086(.248) | 6.546(.191) | 9.307(.249) |
| Run time (in seconds) | 7.2(1.2) | 0.4(.1) | 3.1(.6) | 220.0(30.2) |

LRGC confidence intervals for more challenging high rank data achieves the same empirical coverage rates as that for low rank data. The longer interval is due to the expanding marginal transformation $g_j(z) = z^3$. While all other confidence intervals have visually good coverage rates, their interval lengths are much larger than LRMC confidence intervals, which limits utility.

**MovieLens 1M dataset** We sample the subset of the MovieLens1M data [20] consisting of 2514 movies with at least 50 ratings from 6040 users. We use 80% of observation as training set, 10% as validation set, and 10% as test set, repeated 5 times. On a laptop with Intel-i5-3.1GHz Core and 8 GB RAM, LRGC (rank 10) takes 38 mins in R, softImpute (rank 201) takes 93 mins in R, and GLRM-BvS (rank 200) takes 25 mins in julia. MI-PCA cannot finish even a single imputation in 3 hours in R. We plot the imputation error versus reliability in Figure 2. The value at $m = 100$ is the overall imputation error. All methods have similar overall error. The variance based reliability with GLRM-BvS cannot predict imputation accuracy. In practice, collaborative filtering methods usually recommends very few entries to users. In this setting, LRGC reliability predicts imputation accuracy much better than variance based reliability with softImpute.

**Conclusion** This paper develops a low rank Gaussian copula for matrix completion and quantifies the uncertainty of the resulting imputations. Numerical results show the superiority of our imputation algorithm and the success of our uncertainty measure for predicting imputation accuracy. Quantifying imputation uncertainty can improve algorithms for recommender systems, provide accurate confidence intervals for analyses of scientific and survey data, and enable new active learning strategies.

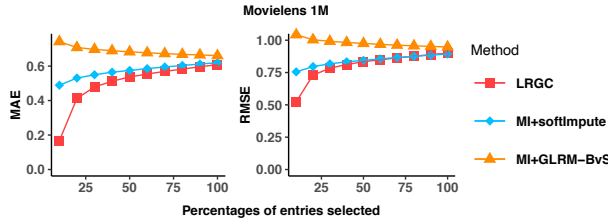

Figure 2: Imputation error on the subset of $m\%$ entries for which method's associated uncertainty metric indicates highest reliability, reported over 5 repetitions (error bars almost invisible).

## Broader Impact

In principle, this paper may benefit any research or practical projects that requires imputing missing values, especially on large scale datasets. The quantified uncertainty may serve as a step to exclude unreliable imputed entries or provide confidence levels after imputation. In areas in which unreliable imputation could have adverse effect on people's lives, such as healthcare datasets, quantified uncertainty is important to aid related decision making. On the other hand, if model assumptions are not met and the resulting quantified uncertainty estimate is not accurate, it may incorrectly lead a practitioner to trust certain entries and distrust others entries when the opposite is true. This problem may be mitigated by running some experiments on a validation set to check whether the proposed method works well on the particular dataset.

## Acknowledgement

We gratefully acknowledge support from NSF Awards IIS-1943131 and CCF-1740822, the ONR Young Investigator Program, DARPA Award FA8750-17-2-0101, the Simons Institute, Canadian Institutes of Health Research, and Capital One. We thank Xin Bing, David Bindel and Thorsten Joachims for helpful discussions. Special thanks go to Xiaoyi Zhu for help in producing our figures.

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
