[Supplementary Material]

# Supplement: Matrix Completion with Quantified Uncertainty through Low Rank Gaussian Copula

**Yuxuan Zhao**
Cornell University
yz2295@cornell.edu

**Madeleine Udell**
Cornell University
udell@cornell.edu

## 1 Proofs

**Setup** Suppose a $p$-dimensional vector $\mathbf{x} \sim \mathrm{LRGC}(\mathbf{W}, \sigma^2, \mathbf{g})$ is observed at locations $\mathcal{O} \subset [p]$ and missing at $\mathcal{M} = [p]/\mathcal{O}$. Then according to the definition of LRGC, for $\mathbf{t} \sim \mathcal{N}(\mathbf{0}, \mathrm{I}_k)$, $\boldsymbol{\epsilon} \sim \mathcal{N}(\mathbf{0}, \sigma^2 \mathrm{I}_p)$, and $\mathbf{z} = \mathbf{W}\mathbf{t} + \boldsymbol{\epsilon}$, we know $\mathbf{x} = \mathbf{g}(\mathbf{z})$ and $\mathbf{z} \sim \mathcal{N}(\mathbf{0}, \boldsymbol{\Sigma})$ with $\boldsymbol{\Sigma} = \mathbf{W}\mathbf{W}^\intercal + \sigma^2 \mathrm{I}_p$. Here we say two random vectors are equal if they have the same CDF.

A key fact we use is that conditional on known $\mathbf{z}_\mathcal{O}$, $\mathbf{z}_\mathcal{M}$ has a normal distribution:

$$\mathbf{z}_\mathcal{M}|\mathbf{z}_\mathcal{O} \sim \mathcal{N}(\boldsymbol{\Sigma}_{\mathcal{M},\mathcal{O}}\boldsymbol{\Sigma}_{\mathcal{O},\mathcal{O}}^{-1}\mathbf{z}_\mathcal{O}, \boldsymbol{\Sigma}_{\mathcal{M},\mathcal{M}} - \boldsymbol{\Sigma}_{\mathcal{M},\mathcal{O}}\boldsymbol{\Sigma}_{\mathcal{O},\mathcal{O}}^{-1}\boldsymbol{\Sigma}_{\mathcal{O},\mathcal{M}}). \tag{1}$$

Here we use $\boldsymbol{\Sigma}_{I,J}$ to denote the submatrix of $\boldsymbol{\Sigma}$ with rows in $I$ and columns in $J$. Plugging in $\boldsymbol{\Sigma} = \mathbf{W}\mathbf{W}^\intercal + \sigma^2 \mathrm{I}_p$, we obtain

$$\begin{aligned}
\mathrm{E}[\mathbf{z}_\mathcal{M}|\mathbf{z}_\mathcal{O}] &= \mathbf{W}_\mathcal{M}\mathbf{W}_\mathcal{O}^\intercal(\mathbf{W}_\mathcal{O}\mathbf{W}_\mathcal{O}^\intercal + \sigma^2\mathbf{I})^{-1}\mathbf{z}_\mathcal{O} \\
&= \mathbf{W}_\mathcal{M}(\sigma^2\mathbf{I} + \mathbf{W}_\mathcal{O}^\intercal\mathbf{W}_\mathcal{O})^{-1}\mathbf{W}_\mathcal{O}^\intercal\mathbf{z}_\mathcal{O}.
\end{aligned} \tag{2}$$

In last equation, we use the Woodbury matrix identity. Similarly, we obtain:

$$\mathrm{Cov}[\mathbf{z}_\mathcal{M}|\mathbf{z}_\mathcal{O}] = \sigma^2\mathbf{I} + \sigma^2\mathbf{W}_\mathcal{M}(\sigma^2\mathbf{I} + \mathbf{W}_\mathcal{O}^\intercal\mathbf{W}_\mathcal{O})^{-1}\mathbf{W}_\mathcal{M}^\intercal. \tag{3}$$

### 1.1 Proof for Lemma 1

Using the law of total expectation,

$$\begin{aligned}
\mathrm{E}[\mathbf{z}_\mathcal{M}|\mathbf{x}_\mathcal{O}] &= \mathrm{E}\left[\mathrm{E}[\mathbf{z}_\mathcal{M}|\mathbf{z}_\mathcal{O}]\big|\mathbf{x}_\mathcal{O}\right] \\
&= \mathrm{E}[\mathbf{W}_\mathcal{M}(\sigma^2\mathbf{I} + \mathbf{W}_\mathcal{O}^\intercal\mathbf{W}_\mathcal{O})^{-1}\mathbf{W}_\mathcal{O}^\intercal\mathbf{z}_\mathcal{O}|\mathbf{x}_\mathcal{O}] \\
&= \mathbf{W}_\mathcal{M}(\sigma^2\mathbf{I} + \mathbf{W}_\mathcal{O}^\intercal\mathbf{W}_\mathcal{O})^{-1}\mathbf{W}_\mathcal{O}^\intercal\mathrm{E}[\mathbf{z}_\mathcal{O}|\mathbf{x}_\mathcal{O}].
\end{aligned}$$

For the first equality, we use Eq. (2).

Similarly we can compute the second moments,

$$\begin{aligned}
\mathrm{E}[\mathbf{z}_\mathcal{M}\mathbf{z}_\mathcal{M}^\intercal|\mathbf{x}_\mathcal{O}] &= \mathrm{E}\left[\mathrm{E}[\mathbf{z}_\mathcal{M}\mathbf{z}_\mathcal{M}^\intercal|\mathbf{z}_\mathcal{O}]\big|\mathbf{x}_\mathcal{O}\right] \\
&= \mathrm{E}\left[\mathrm{E}[\mathbf{z}_\mathcal{M}|\mathbf{z}_\mathcal{O}]\mathrm{E}[\mathbf{z}_\mathcal{M}|\mathbf{z}_\mathcal{O}]^\intercal + \mathrm{Cov}[\mathbf{z}_\mathcal{M}|\mathbf{z}_\mathcal{O}]|\mathbf{x}_\mathcal{O}\right] \\
&= \mathrm{E}\left[\mathrm{E}[\mathbf{z}_\mathcal{M}|\mathbf{z}_\mathcal{O}]\mathrm{E}[\mathbf{z}_\mathcal{M}|\mathbf{z}_\mathcal{O}]^\intercal|\mathbf{x}_\mathcal{O}\right] + \mathrm{E}\left[\mathrm{Cov}[\mathbf{z}_\mathcal{M}|\mathbf{z}_\mathcal{O}]|\mathbf{x}_\mathcal{O}\right] \\
&= \mathrm{E}\left[\mathrm{E}[\mathbf{z}_\mathcal{M}|\mathbf{z}_\mathcal{O}]\mathrm{E}[\mathbf{z}_\mathcal{M}|\mathbf{z}_\mathcal{O}]^\intercal|\mathbf{x}_\mathcal{O}\right] + \mathrm{Cov}[\mathbf{z}_\mathcal{M}|\mathbf{z}_\mathcal{O}].
\end{aligned} \tag{4}$$

From the last equation, we use the fact that $\mathrm{Cov}[\mathbf{z}_\mathcal{M}|\mathbf{x}_\mathcal{O}]$ is fully determined by $\mathbf{W}$ and $\sigma^2$ and thus does not depend on $\mathbf{x}_\mathcal{M}$.

Plug Eq. (2) and Eq. (3) into Eq. (4) to obtain

$$\begin{aligned}
\mathrm{E}\left[\mathrm{E}[\mathbf{z}_\mathcal{M}|\mathbf{z}_\mathcal{O}]\mathrm{E}[\mathbf{z}_\mathcal{M}|\mathbf{z}_\mathcal{O}]^\intercal|\mathbf{x}_\mathcal{O}\right] = \\
\mathbf{W}_\mathcal{M}(\sigma^2\mathbf{I} + \mathbf{W}_\mathcal{O}^\intercal\mathbf{W}_\mathcal{O})^{-1}\mathbf{W}_\mathcal{O}^\intercal\mathrm{E}[\mathbf{z}_\mathcal{O}\mathbf{z}_\mathcal{O}^\intercal|\mathbf{x}_\mathcal{O}]\mathbf{W}_\mathcal{O}(\sigma^2\mathbf{I} + \mathbf{W}_\mathcal{O}^\intercal\mathbf{W}_\mathcal{O})^{-1}\mathbf{W}_\mathcal{M}^\intercal.
\end{aligned}$$

Then using $\mathrm{Cov}[\mathbf{z}_{\mathcal{M}}|\mathbf{x}_{\mathcal{O}}] = \mathrm{E}[\mathbf{z}_{\mathcal{M}}\mathbf{z}_{\mathcal{M}}^{\mathsf{T}}|\mathbf{x}_{\mathcal{O}}] - \mathrm{E}[\mathbf{z}_{\mathcal{M}}|\mathbf{x}_{\mathcal{O}}]\mathrm{E}[\mathbf{z}_{\mathcal{M}}^{\mathsf{T}}|\mathbf{x}_{\mathcal{O}}]$, we have

$$\mathrm{Cov}[\mathbf{z}_{\mathcal{M}}|\mathbf{x}_{\mathcal{O}}] = \sigma^2 \mathbf{I}_{|\mathcal{M}|} + \sigma^2 \mathbf{W}_{\mathcal{M}}(\sigma^2\mathbf{I} + \mathbf{W}_{\mathcal{O}}^{\mathsf{T}}\mathbf{W}_{\mathcal{O}})^{-1}\mathbf{W}_{\mathcal{M}}^{\mathsf{T}} +$$
$$\mathbf{W}_{\mathcal{M}}(\sigma^2\mathbf{I} + \mathbf{W}_{\mathcal{O}}^{\mathsf{T}}\mathbf{W}_{\mathcal{O}})^{-1}\mathbf{W}_{\mathcal{O}}^{\mathsf{T}}\mathrm{Cov}[\mathbf{z}_{\mathcal{O}}|\mathbf{x}_{\mathcal{O}}]\mathbf{W}_{\mathcal{O}}(\sigma^2\mathbf{I} + \mathbf{W}_{\mathcal{O}}^{\mathsf{T}}\mathbf{W}_{\mathcal{O}})^{-1}\mathbf{W}_{\mathcal{M}}^{\mathsf{T}}.$$

## 1.2 Proof of Theorem 1

*Proof.* Theorem 1 is an immediate consequence of the normality of $\mathbf{z}_{\mathcal{M}}$ conditional on $\mathbf{z}_{\mathcal{O}} = \mathbf{g}_{\mathcal{O}}^{-1}(\mathbf{x}_{\mathcal{O}})$ (see Eq. (1)) and the elementwise strictly monotone $\mathbf{g}$. $\qquad\square$

## 1.3 Proof of Theorem 2

Suppose $\mathbf{x} = (x_1, \ldots, x_p)$ where $x_j$ is ordinal with $k_j(\geq 2)$ ordinal levels encoded as $\{1, \ldots, k_j\}$ for $j \in [p]$. For ordinal data, the conditional distribution of $\mathbf{z}_{\mathcal{M}}|\mathbf{z}_{\mathcal{O}} \in \mathbf{g}_{\mathcal{O}}^{-1}(\mathbf{x}_{\mathcal{O}})$ is intractable. Consequently, we cannot establish distribution-based confidence intervals for $\mathbf{z}_{\mathcal{M}}$.

Instead, for each marginal $j$, we can lower bound the probability of event $|\hat{x}_j - x_j| \leq d$ for the LRGC imputation $\hat{x}_j$ and $d \in \mathbb{Z}$. Since $\mathrm{Pr}(|\hat{x}_j - x_j| \leq k_j - 1) = 1$, it suffices to consider $d < k_j - 1$. In practice, the result is more useful for small $d$, such as $d = 0$. Let us first state a generalization of our Theorem 2.

**Theorem 4.** Suppose $\mathbf{x} \sim \mathrm{LRGC}(\mathbf{W}, \sigma^2, \mathbf{g})$ with observations $\mathbf{x}_{\mathcal{O}}$ and missing entries $\mathbf{x}_{\mathcal{M}}$. Also for each marginal $j \in [p]$, $x_j$ takes values from $\{1, \ldots, k_j\}$ and thus the $g_j$ is a step function with cut points $\mathbf{S}_j = \{s_1, \ldots, s_{k_j-1}\}$:

$$g_j(z) = 1 + \sum_{k=1}^{k_j-1} \mathbb{1}(z > s_k), \quad \text{where} \quad -\infty =: s_0 < s_1 < \ldots < s_{k_j-1} < s_{k_j} := \infty.$$

For a missing entry $x_j$, $j \in \mathcal{M}$, the set of values for $z_j$ that would yield the same imputed value $\hat{x}_j = g_j(\mathrm{E}[z_j|\mathbf{x}_{\mathcal{O}}])$ is $g_j^{-1}(\hat{x}_j) = (s_{\hat{x}_j-1}, s_{\hat{x}_j}]$. Then the following holds:

$$\mathrm{Pr}(|\hat{x}_j - x_j| \leq d) \geq 1 - \frac{\mathrm{Var}[z_j|\mathbf{x}_{\mathcal{O}}]}{d_j^2},$$

with

$$d_j = \min(|\mathrm{E}[z_j|\mathbf{x}_{\mathcal{O}}] - s_{\max(\hat{x}_j-1-d,0)}|, |\mathrm{E}[z_j|\mathbf{x}_{\mathcal{O}}] - s_{\min(\hat{x}_j+d,k_j)}|),$$

where $\mathrm{E}[z_j|\mathbf{x}_{\mathcal{O}}]$, $\mathrm{Var}[z_j|\mathbf{x}_{\mathcal{O}}]$ are given in Lemma 1 with $\mathcal{M}$ replaced by $j$.

*Proof.* The proof applies to each missing dimension $j \in \mathcal{M}$. Let us further define $s_k = -\infty$ for any negative integer $k$ and $s_k = \infty$ for any integer $k > k_j$ for convenience. Then $s_k = s_{\max(k,0)}$ for negative integer $k$ and $s_k = s_{\min(k,k_j)}$ for integer $k$ larger than $k_j$.

First notice $|\hat{x}_j - x_j| \leq d$ if and only if $z_j \in (s_{\hat{x}_j-1-d}, s_{\hat{x}_j+d}]$ for the latent normal $z_j$ satisfying $x_j = g_j(z_j)$. Specifically, when $d = 0$, $\hat{x}_j = x_j$ if and only if $z_j \in (s_{\hat{x}_j-1}, s_{\hat{x}_j}]$, i.e. $g_j^{-1}(x_j) = (s_{\hat{x}_j-1}, s_{\hat{x}_j}] = g_j^{-1}(\hat{x}_j)$. Notice we have,

$$\mathrm{E}[z_j|\mathbf{x}_{\mathcal{O}}] \in (s_{\hat{x}_j-1}, s_{\hat{x}_j}] \subset (s_{\hat{x}_j-1-d}, s_{\hat{x}_j+d}].$$

Thus a sufficient condition for $z_j \in (s_{\hat{x}_j-1-d}, s_{\hat{x}_j+d}]$ is that $z_j$ is sufficiently close to its conditional mean $\mathrm{E}[z_j|\mathbf{x}_{\mathcal{O}}]$. More precisely,

$$|\mathrm{E}[z_j|\mathbf{x}_{\mathcal{O}}] - z_j| \leq \min(|\mathrm{E}[z_j|\mathbf{x}_{\mathcal{O}}] - s_{\hat{x}_j-1-d}|, |\mathrm{E}[z_j|\mathbf{x}_{\mathcal{O}}] - s_{\hat{x}_j+d}|) \to |\hat{x}_j - x_j| \leq d.$$

Define $d_j := \min(|\mathrm{E}[z_j|\mathbf{x}_{\mathcal{O}}] - s_{\hat{x}_j-1-d}|, |\mathrm{E}[z_j|\mathbf{x}_{\mathcal{O}}] - s_{\hat{x}_j+d}|)$. Notice when $d = 0$,

$$d_j = \min(|\mathrm{E}[z_j|\mathbf{x}_{\mathcal{O}}] - s_{\hat{x}_j-1}|, |\mathrm{E}[z_j|\mathbf{x}_{\mathcal{O}}] - s_{\hat{x}_j}|) = \min_{s \in \mathbf{S}} |\mathrm{E}[z_j|\mathbf{x}_{\mathcal{O}}] - s|.$$

Use the Markov inequality together with the conditional distribution of $z_j$ given $\mathbf{x}_{\mathcal{O}}$ to bound

$$\mathrm{Pr}(|\mathrm{E}[z_j|\mathbf{x}_{\mathcal{O}}] - z_j| > d_j) \leq \frac{\mathrm{Var}[z_j|\mathbf{x}_{\mathcal{O}}]}{d_j^2},$$

which completes our proof. $\qquad\square$

## 1.4 Proof of Theorem 3

To prove Theorem 3, we introduce a lemma which provides a concentration inequality on quadratic forms of sub-Gaussian vectors. For a detailed treatment of sub-Gaussian random distributions, see [10]. A random variable $x \in \mathbb{R}$ is called sub-Gaussian if $(\mathrm{E}[|x|^p])^{1/p} \leq K\sqrt{p}$ for all $p \geq 1$ with some $K > 0$. The sub-Gaussian norm of $x$ is defined as $||x||_{\psi_2} = \sup_{p \geq 1} p^{-1/2} (\mathrm{E}[|x|^p])^{1/p}$.

Denote the inner product of vectors $\mathbf{x}_1$ and $\mathbf{x}_2$ as $\langle \mathbf{x}_1, \mathbf{x}_2 \rangle$. A random vector $\mathbf{x} \in \mathbb{R}^p$ is called sub-Gaussian if the one-dimensional marginals $\langle \mathbf{x}, \mathbf{a} \rangle$ are all sub-Gaussian random variables for any constant vector $\mathbf{a} \in \mathbb{R}^p$. The sub-Gaussian norm of $\mathbf{x}$ is defined as $||\mathbf{x}||_{\psi_2} = \sup_{\mathbf{a} \in \mathbb{S}^{p-1}} ||\langle \mathbf{x}, \mathbf{a} \rangle||_{\psi_2}$. A Gaussian random vector is also sub-Gaussian.

**Lemma 2.** Let $\Sigma \in \mathbb{R}^{p \times p}$ be a positive semidefinite matrix. Let $\mathbf{x} = (x_1, \ldots, x_p)$ be a sub-Gaussian random vector with mean zero and covariance matrix $\mathbf{I}_p$. For all $t > 0$,

$$\Pr \left[ \mathbf{x}^{\mathsf{T}} \Sigma \mathbf{x} > (\sqrt{\mathrm{tr}(\Sigma)} + \sqrt{2\lambda_1(\Sigma)t})^2 \right] \leq e^{-t}.$$

Our Lemma 2 is Lemma 17 in [1], which is also a simplified version of Theorem 1 in [4].

*Proof.* Since $\mathbf{g}$ is elementwise Lipschitz with constant $L$,

$$\mathrm{MSE}(\hat{\mathbf{x}}) = \frac{||\mathbf{g}_{\mathcal{M}}(\mathbf{z}_{\mathcal{M}}) - \mathbf{g}_{\mathcal{M}}(\hat{\mathbf{z}}_{\mathcal{M}})||_2^2}{||\mathcal{M}||} \leq L^2 \frac{||\mathbf{z}_{\mathcal{M}} - \hat{\mathbf{z}}_{\mathcal{M}}||_2^2}{||\mathcal{M}||}. \tag{5}$$

Denote the covariance matrix of $\mathbf{z}_{\mathcal{M}}$ conditional on $\mathbf{z}_{\mathcal{O}}$ as $\Sigma_{(\mathcal{M})}$. Apply the above inequality with $\Sigma = \Sigma_{(\mathcal{M})}$ and $\mathbf{x} = \Sigma_{(\mathcal{M})}^{-1/2} \mathbf{z}_{\mathcal{M}}$, we obtain:

$$\Pr \left( ||\mathbf{z}_{\mathcal{M}} - \hat{\mathbf{z}}_{\mathcal{M}}||_2^2 > \left( \sqrt{\mathrm{tr}(\Sigma_{(\mathcal{M})})} + \sqrt{2\lambda_1(\Sigma_{(\mathcal{M})})t} \right)^2 \right) \leq e^{-t}. \tag{6}$$

Notice

$$\begin{aligned}
\mathrm{tr}(\Sigma_{(\mathcal{M})}) &= \mathrm{tr} \left( \sigma^2 \mathbf{I} + \sigma^2 \mathbf{W}_{\mathcal{M}} (\sigma^2 \mathbf{I} + \mathbf{W}_{\mathcal{O}}^{\mathsf{T}} \mathbf{W}_{\mathcal{O}})^{-1} \mathbf{W}_{\mathcal{M}}^{\mathsf{T}} \right) \\
&= \sigma^2 |\mathcal{M}| + \sigma^2 \mathrm{tr} \left( (\sigma^2 \mathbf{I} + \mathbf{W}_{\mathcal{O}}^{\mathsf{T}} \mathbf{W}_{\mathcal{O}})^{-1} \mathbf{W}_{\mathcal{M}}^{\mathsf{T}} \mathbf{W}_{\mathcal{M}} \right) \\
&\leq \sigma^2 |\mathcal{M}| + \sigma^2 \lambda_1(\sigma^2 \mathbf{I} + \mathbf{W}_{\mathcal{O}}^{\mathsf{T}} \mathbf{W}_{\mathcal{O}})^{-1}) \mathrm{tr} \left( \mathbf{W}_{\mathcal{M}}^{\mathsf{T}} \mathbf{W}_{\mathcal{M}} \right) \\
&= \sigma^2 |\mathcal{M}| + \sigma^2 \frac{1}{\sigma^2 + \lambda_k^2(\mathbf{W}_{\mathcal{O}})} (1 - \sigma^2) |\mathcal{M}|.
\end{aligned}$$

In the inequality, we use the fact $\mathrm{tr}(\mathbf{AB}) \leq \lambda_1(\mathbf{A}) \mathrm{tr}(\mathbf{B})$ for any real symmetric positive semidefinite matrices $\mathbf{A}$ and $\mathbf{B}$. In the last equation, we use the unit diagonal constraints of $\mathbf{WW}^{\mathsf{T}} + \sigma^2 \mathbf{I}_p$ such that $\mathrm{tr}(\mathbf{W}_{\mathcal{M}}^{\mathsf{T}} \mathbf{W}_{\mathcal{M}}) = \mathrm{tr}(\mathbf{W}_{\mathcal{M}} \mathbf{W}_{\mathcal{M}}^{\mathsf{T}}) = |\mathcal{M}|(1 - \sigma^2)$.

Also notice

$$\begin{aligned}
\lambda_1(\Sigma_{(\mathcal{M})}) &= \lambda_1(\sigma^2 \mathbf{I} + \sigma^2 \mathbf{W}_{\mathcal{M}} (\sigma^2 \mathbf{I} + \mathbf{W}_{\mathcal{O}}^{\mathsf{T}} \mathbf{W}_{\mathcal{O}})^{-1} \mathbf{W}_{\mathcal{M}}^{\mathsf{T}}) \\
&\leq \sigma^2 + \sigma^2 \lambda_1(\mathbf{W}_{\mathcal{M}} (\sigma^2 \mathbf{I} + \mathbf{W}_{\mathcal{O}}^{\mathsf{T}} \mathbf{W}_{\mathcal{O}})^{-1} \mathbf{W}_{\mathcal{M}}^{\mathsf{T}}) \\
&\leq \sigma^2 + \sigma^2 \lambda_1^2(\mathbf{W}_{\mathcal{M}}) \lambda_1((\sigma^2 \mathbf{I} + \mathbf{W}_{\mathcal{O}}^{\mathsf{T}} \mathbf{W}_{\mathcal{O}})^{-1}) \\
&= \sigma^2 + \sigma^2 \frac{\lambda_1^2(\mathbf{W}_{\mathcal{M}})}{\sigma^2 + \lambda_k^2(\mathbf{W}_{\mathcal{O}})}.
\end{aligned}$$

Thus,

$$||\mathbf{z}_{\mathcal{M}} - \hat{\mathbf{z}}_{\mathcal{M}}||_2^2 \leq \sigma^2 |\mathcal{M}| \cdot \left( \sqrt{1 + \frac{1 - \sigma^2}{\sigma^2 + \lambda_k^2(\mathbf{W}_{\mathcal{O}})}} + \sqrt{\left( 1 + \frac{\lambda_1^2(\mathbf{W}_{\mathcal{M}})}{\sigma^2 + \lambda_k^2(\mathbf{W}_{\mathcal{O}})} \right) \frac{2t}{|\mathcal{M}|}} \right)^2. \tag{7}$$

Combining Eq. (5), Eq. (6) and Eq. (7), we finish the proof. $\square$

## 1.5   Proof of Corollary 1

We first introduce a result from [10, Theorem 5.39] characterizing the singular values of long random matrices with independent sub-Gaussian rows.

**Lemma 3.** Let $\mathbf{A} \in \mathbb{R}^{p \times k}$ be a matrix whose rows $\mathbf{a}_j$ are independent sub-Gaussian random vectors in $\mathbb{R}^k$ whose covariance matrix is $\mathbf{\Sigma}$. Then for every $t > 0$, with probability as least $1 - 2\exp(-ct^2)$ one has

$$\lambda_1 \left( \frac{1}{p} \mathbf{A}^\intercal \mathbf{A} - \mathbf{\Sigma} \right) \leq \max(\delta, \delta^2) \lambda_1(\mathbf{\Sigma}), \quad \text{where } \delta = C\sqrt{\frac{k}{p}} + \frac{t}{\sqrt{p}}.$$

Here $c, C > 0$ depend only on the subgaussian norm $K = \max_j ||\mathbf{\Sigma}^{-1/2} \mathbf{a}_j||_{\psi_2}$.

*Proof.* Apply Lemma 3 to submatrix $\mathbf{W}_{\mathcal{O}}$ and $\mathbf{W}_{\mathcal{M}}$ respectively with covariance matrix $\mathbf{\Sigma} = \frac{1-\sigma^2}{k} \mathbf{I}_k$, we obtain with probability at least $1 - 2\exp(-ct_1^2) - 2\exp(-ct_2^2)$,

$$\left| \frac{1}{|\mathcal{O}|} \lambda_k^2(\mathbf{W}_{\mathcal{O}}) - \frac{1-\sigma^2}{k} \right| \leq \frac{1-\sigma^2}{k} \epsilon_1 \text{ and } \left| \frac{1}{|\mathcal{M}|} \lambda_1^2(\mathbf{W}_{\mathcal{M}}) - \frac{1-\sigma^2}{k} \right| \leq \frac{(1-\sigma^2)\epsilon_2}{k},$$

where $\epsilon_1 = \max(\delta_1, \delta_1^2)$ with $\delta_1 = \frac{C\sqrt{k}+t_1}{\sqrt{|\mathcal{O}|}}$ and $\epsilon_2 = \max(\delta_2, \delta_2^2)$ with $\delta_2 = \frac{C\sqrt{k}+t_2}{\sqrt{|\mathcal{M}|}}$. Constants $c, C > 0$ only depend on the subgaussian norm $\max_j ||\sqrt{\frac{k}{1-\sigma^2}} \mathbf{w}_j||_{\psi_2}$.

For any $0 < \epsilon < 1$, let $t_1 = \frac{\epsilon\sqrt{|\mathcal{O}|}}{2}$ and $t_2 = \frac{\epsilon\sqrt{|\mathcal{O}|}}{2\sqrt{c_2}}$. Suppose the sufficiently large constant $c_1$ satisfies $c_1 > \frac{4C^2 \max(1, c_2)}{\epsilon^2}$. Then we have

$$\epsilon_1 = \delta_1 = \frac{C}{\sqrt{|\mathcal{O}|/k}} + \frac{t}{\sqrt{|\mathcal{O}|}} < \frac{C}{\sqrt{c_1}} + \frac{\epsilon}{2} < \frac{C}{\sqrt{4C^2/\epsilon^2}} + \frac{\epsilon}{2} = \epsilon,$$

and

$$\epsilon_2 = \delta_2 = \frac{C}{\sqrt{|\mathcal{M}|/k}} + \frac{t}{\sqrt{|\mathcal{M}|}} < \frac{C}{\sqrt{|\mathcal{O}|/c_2 k}} + \frac{\epsilon}{2} < \frac{C}{\sqrt{4C^2/\epsilon^2}} + \frac{\epsilon}{2} = \epsilon.$$

Thus we have with probability at least $1 - 2\exp(-c\epsilon^2 |\mathcal{O}|/4) - 2\exp(-c\epsilon^2 |\mathcal{O}|/4c_2)$,

$$\lambda_k^2(\mathbf{W}_{\mathcal{O}}) > (1-\sigma^2)(1-\epsilon)\frac{|\mathcal{O}|}{k} \quad \text{and} \quad \lambda_1^2(\mathbf{W}_{\mathcal{M}}) \leq (1-\sigma^2)(1+\epsilon)\frac{|\mathcal{M}|}{k}. \tag{8}$$

Combining Eq. (7) and Eq. (8), then with probability at least $1 - \exp(-t) - 2\exp(-c\epsilon^2|\mathcal{O}|/4) - 2\exp(-c\epsilon^2|\mathcal{O}|/4c_2)$,

$$\frac{||\mathbf{z}_{\mathcal{M}} - \hat{\mathbf{z}}_{\mathcal{M}}||_2^2}{|\mathcal{M}|} \leq \sigma^2 \left( \sqrt{1 + \frac{1}{\frac{\sigma^2}{1-\sigma^2} + (1-\epsilon)|\mathcal{O}|/k}} + \sqrt{\frac{2t}{|\mathcal{M}|} + \frac{2(1+\epsilon)t}{\frac{k\sigma^2}{1-\sigma^2} + (1-\epsilon)|\mathcal{O}|}} \right)^2$$

$$\leq \sigma^2 \left( \sqrt{1 + \frac{1}{\frac{\sigma^2}{1-\sigma^2} + (1-\epsilon)|\mathcal{O}|/k}} + \sqrt{\frac{2c_2 t}{|\mathcal{O}|} + \frac{2(1+\epsilon)t}{\frac{k\sigma^2}{1-\sigma^2} + (1-\epsilon)|\mathcal{O}|}} \right)^2. \tag{9}$$

Now take $t = \log|\mathcal{O}|$, with fixed $k$ and $\sigma^2$, the right hand side is $1 + O\left( \sqrt{\frac{\log|\mathcal{O}|}{|\mathcal{O}|}} \right)$.

Notice $|\mathcal{O}| > c_1 k \geq c_1$. Then there exists some constant $c_3 > 0$ such that $|\mathcal{O}|$ satisfies:

$$\log|\mathcal{O}| < c_3 \frac{c\epsilon^2 |\mathcal{O}|}{4 \max(1, c_2)}$$

thus Eq. (9) holds with probability at least $1 - \frac{1+2c_3}{|\mathcal{O}|}$. Combing the result with Eq. (5) completes the proof. $\qquad\square$

## 2 Algorithm detail

### 2.1 E-step details

We provide details on E-step computation here. The key fact we use is that conditional on known $\mathbf{z}_{\mathcal{O}}$, $\mathbf{t}$ is normally distributed:

$$\mathbf{t}|\mathbf{z}_{\mathcal{O}} \sim \mathcal{N}(\mathbf{M}_{\mathcal{O}}^{-1}\mathbf{W}_{\mathcal{O}}^{\mathsf{T}}\mathbf{z}_{\mathcal{O}}, \sigma^2\mathbf{M}_{\mathcal{O}}^{-1}), \tag{10}$$

where $\mathbf{M}_{\mathcal{O}} = \sigma^2\mathbf{I}_k + \mathbf{W}_{\mathcal{O}}^{\mathsf{T}}\mathbf{W}_{\mathcal{O}}$. This result follows by applying the Bayes formula with $\mathbf{z}_{\mathcal{O}}|\mathbf{t} \sim \mathcal{N}(\mathbf{W}_{\mathcal{O}}\mathbf{t}, \sigma^2\mathbf{I}_p), \mathbf{t} \sim \mathcal{N}(\mathbf{0}, \mathbf{I}_k)$ and $\mathbf{z}_{\mathcal{O}} \sim \mathcal{N}(\mathbf{0}, \mathbf{W}_{\mathcal{O}}\mathbf{W}_{\mathcal{O}}^{\mathsf{T}} + \sigma^2\mathbf{I}_p)$.

First we express the Q-function $Q(Q(\mathbf{W}, \sigma^2; \tilde{\mathbf{W}}, \tilde{\sigma}^2))$:

$$Q(\mathbf{W}, \sigma^2; \tilde{\mathbf{W}}, \tilde{\sigma}^2) = c - \frac{\sum_{i=1}^{n}|\mathcal{O}_i|\log(\sigma^2)}{2}$$
$$- \frac{\sum_{i=1}^{n}\left(\mathbb{E}[(\mathbf{z}_{\mathcal{O}_i}^i)^{\mathsf{T}}\mathbf{z}_{\mathcal{O}_i}^i] - 2\mathrm{tr}(\mathbf{W}_{\mathcal{O}_i}\mathbb{E}[\mathbf{t}^i(\mathbf{z}_{\mathcal{O}_i}^i)^{\mathsf{T}}]) + \mathrm{tr}(\mathbf{W}_{\mathcal{O}_i}^{\mathsf{T}}\mathbf{W}_{\mathcal{O}_i}\mathbb{E}[\mathbf{t}^i(\mathbf{t}^i)^{\mathsf{T}}])\right)}{2\sigma^2}, \tag{11}$$

where $c$ is an absolute constant in terms the model parameters $\mathbf{W}$ and $\sigma^2$.

Thus to evaluate the $Q$ function, we only need (1)$\mathbb{E}[(\mathbf{z}_{\mathcal{O}_i}^i)^{\mathsf{T}}\mathbf{z}_{\mathcal{O}_i}^i]$, (2)$\mathbb{E}[\mathbf{t}^i(\mathbf{z}_{\mathcal{O}_i}^i)^{\mathsf{T}}]$ and (3)$\mathbb{E}[\mathbf{t}^i(\mathbf{t}^i)^{\mathsf{T}}]$. computing (1) only needs $\mathrm{E}[\mathbf{z}_{\mathcal{O}_i}^i|\mathbf{x}_{\mathcal{O}_i}^i]$ and $\mathrm{Cov}[\mathbf{z}_{\mathcal{O}_i}^i|\mathbf{x}_{\mathcal{O}_i}^i]$. To compute (2) and (3), we use the law of total expectation similar as in Section 1.1 by first treating $\mathbf{z}_{\mathcal{O}_i}^i$ as known.

Since $\mathrm{E}[\mathbf{t}^i|\mathbf{z}_{\mathcal{O}_i}^i] = \mathbf{M}_{\mathcal{O}_i}^{-1}\mathbf{W}_{\mathcal{O}_i}^{\mathsf{T}}\mathbf{z}_{\mathcal{O}_i}^i$ and $\mathrm{Cov}[\mathbf{t}^i|\mathbf{z}_{\mathcal{O}_i}^i] = \sigma^2\mathbf{M}_{\mathcal{O}_i}^{-1}$, we have

$$\mathrm{E}[\mathbf{t}^i|\mathbf{x}_{\mathcal{O}_i}^i] = \mathrm{E}[\mathrm{E}[\mathbf{t}^i|\mathbf{z}_{\mathcal{O}_i}^i]|\mathbf{x}_{\mathcal{O}_i}^i]$$
$$= \mathrm{E}[\mathbf{M}_{\mathcal{O}_i}^{-1}\mathbf{W}_{\mathcal{O}_i}^{\mathsf{T}}\mathbf{z}_{\mathcal{O}_i}^i|\mathbf{x}_{\mathcal{O}_i}^i]$$
$$= \mathbf{M}_{\mathcal{O}_i}^{-1}\mathbf{W}_{\mathcal{O}_i}^{\mathsf{T}}\mathrm{E}[\mathbf{z}_{\mathcal{O}_i}^i|\mathbf{x}_{\mathcal{O}_i}^i].$$

Then

$$\mathrm{E}[\mathbf{t}^i(\mathbf{z}_{\mathcal{O}_i}^i)^{\mathsf{T}}|\mathbf{x}_{\mathcal{O}_i}^i] = \mathrm{E}[\mathrm{E}[\mathbf{t}^i(\mathbf{z}_{\mathcal{O}_i}^i)^{\mathsf{T}}|\mathbf{z}_{\mathcal{O}_i}^i]|\mathbf{x}_{\mathcal{O}_i}^i]$$
$$= \mathrm{E}[\mathrm{E}[\mathbf{t}^i|\mathbf{z}_{\mathcal{O}_i}^i](\mathbf{z}_{\mathcal{O}_i}^i)^{\mathsf{T}}|\mathbf{x}_{\mathcal{O}_i}^i]$$
$$= \mathbf{M}_{\mathcal{O}_i}^{-1}\mathbf{W}_{\mathcal{O}_i}^{\mathsf{T}}\mathrm{E}[\mathbf{z}_{\mathcal{O}_i}^i(\mathbf{z}_{\mathcal{O}_i}^i)^{\mathsf{T}}|\mathbf{x}_{\mathcal{O}_i}^i]$$
$$= \mathbf{M}_{\mathcal{O}_i}^{-1}\mathbf{W}_{\mathcal{O}_i}^{\mathsf{T}}\left(\mathrm{Cov}[\mathbf{z}_{\mathcal{O}_i}^i|\mathbf{x}_{\mathcal{O}_i}^i] + \mathrm{E}[\mathbf{z}_{\mathcal{O}_i}^i|\mathbf{x}_{\mathcal{O}_i}^i]\mathrm{E}[(\mathbf{z}_{\mathcal{O}_i}^i)^{\mathsf{T}}|\mathbf{x}_{\mathcal{O}_i}^i]\right)$$
$$= \mathbf{M}_{\mathcal{O}_i}^{-1}\mathbf{W}_{\mathcal{O}_i}^{\mathsf{T}}\mathrm{Cov}[\mathbf{z}_{\mathcal{O}_i}^i|\mathbf{x}_{\mathcal{O}_i}^i] + \mathrm{E}[\mathbf{t}^i|\mathbf{x}_{\mathcal{O}_i}^i]\mathrm{E}[(\mathbf{z}_{\mathcal{O}_i}^i)^{\mathsf{T}}|\mathbf{x}_{\mathcal{O}_i}^i].$$

and

$$\mathrm{E}[\mathbf{t}^i(\mathbf{t}^i)^{\mathsf{T}}|\mathbf{x}_{\mathcal{O}_i}^i] = \mathrm{E}[\mathrm{E}[\mathbf{t}^i(\mathbf{t}^i)^{\mathsf{T}}|\mathbf{z}_{\mathcal{O}_i}^i]|\mathbf{x}_{\mathcal{O}_i}^i]$$
$$= \mathrm{E}[\mathbf{M}_{\mathcal{O}_i}^{-1}\mathbf{W}_{\mathcal{O}_i}^{\mathsf{T}}\mathbf{z}_{\mathcal{O}_i}^i(\mathbf{z}_{\mathcal{O}_i}^i)^{\mathsf{T}}\mathbf{W}_{\mathcal{O}_i}\mathbf{M}_{\mathcal{O}_i}^{-1}|\mathbf{x}_{\mathcal{O}_i}^i] + \mathrm{E}[\mathrm{Cov}[\mathbf{t}^i|\mathbf{z}_{\mathcal{O}_i}^i]|\mathbf{x}_{\mathcal{O}_i}^i]$$
$$= \mathbf{M}_{\mathcal{O}_i}^{-1}\mathbf{W}_{\mathcal{O}_i}^{\mathsf{T}}\mathrm{E}[\mathbf{z}_{\mathcal{O}_i}^i(\mathbf{z}_{\mathcal{O}_i}^i)^{\mathsf{T}}|\mathbf{x}_{\mathcal{O}_i}^i]\mathbf{W}_{\mathcal{O}_i}\mathbf{M}_{\mathcal{O}_i}^{-1} + \mathrm{E}[\sigma^2\mathbf{M}_{\mathcal{O}_i}^{-1}|\mathbf{x}_{\mathcal{O}_i}^i]$$
$$= \mathbf{M}_{\mathcal{O}_i}^{-1}\mathbf{W}_{\mathcal{O}_i}^{\mathsf{T}}\left(\mathrm{Cov}[\mathbf{z}_{\mathcal{O}_i}^i|\mathbf{x}_{\mathcal{O}_i}^i] + \mathrm{E}[\mathbf{z}_{\mathcal{O}_i}^i|\mathbf{x}_{\mathcal{O}_i}^i]\mathrm{E}[(\mathbf{z}_{\mathcal{O}_i}^i)^{\mathsf{T}}|\mathbf{x}_{\mathcal{O}_i}^i]\right)\mathbf{W}_{\mathcal{O}_i}\mathbf{M}_{\mathcal{O}_i}^{-1} + \sigma^2\mathbf{M}_{\mathcal{O}_i}^{-1}$$
$$= \mathbf{M}_{\mathcal{O}_i}^{-1}\mathbf{W}_{\mathcal{O}_i}^{\mathsf{T}}\mathrm{Cov}[\mathbf{z}_{\mathcal{O}_i}^i|\mathbf{x}_{\mathcal{O}_i}^i]\mathbf{W}_{\mathcal{O}_i}\mathbf{M}_{\mathcal{O}_i}^{-1} + \mathrm{E}[\mathbf{t}^i|\mathbf{x}_{\mathcal{O}_i}^i]\mathrm{E}[(\mathbf{t}^i)^{\mathsf{T}}|\mathbf{x}_{\mathcal{O}_i}^i] + \sigma^2\mathbf{M}_{\mathcal{O}_i}^{-1}.$$

### 2.2 M-step details

Take the derivative of the Q-function in Eq. (11) with respect to row $\mathbf{w}_j^{\mathsf{T}}$ and $\sigma^2$:

$$\frac{\partial Q}{\partial \mathbf{w}_j^{\mathsf{T}}} = \frac{-1}{|\Omega_j|\sigma^2}\sum_{i\in\Omega_j}(-\mathbf{e}_j^{\mathsf{T}}\mathbb{E}[\mathbf{z}_{\mathcal{O}_i}^i\mathbf{t}_i^{\mathsf{T}}] + \mathbf{w}_j^{\mathsf{T}}\mathbb{E}[\mathbf{t}_i\mathbf{t}_i^{\mathsf{T}}]),$$

$$\frac{\partial Q}{\partial \sigma^2} = \frac{1}{2\sigma^4}\sum_{i=1}^{n}\left(\mathbb{E}[(\mathbf{z}_{\mathcal{O}_i}^i)^{\mathsf{T}}\mathbf{z}_{\mathcal{O}_i}^i] - 2\mathrm{tr}(\mathbf{W}_{\mathcal{O}_i}\mathbb{E}[\mathbf{t}_i(\mathbf{z}_{\mathcal{O}_i}^i)^{\mathsf{T}}]) + \mathrm{tr}(\mathbf{W}_{\mathcal{O}_i}^{\mathsf{T}}\mathbf{W}_{\mathcal{O}_i}\mathbb{E}[\mathbf{t}_i\mathbf{t}_i^{\mathsf{T}}])\right) - \frac{\sum_{i=1}^{n}|\mathcal{O}_i|}{2\sigma^2}.$$

Set both to zero to obtain the update for M-step:

$$\hat{\mathbf{w}}_j^\mathsf{T} = \left( \mathbf{e}_j^\mathsf{T} \sum_{i \in \Omega_j} \mathbb{E}[\mathbf{z}_{\mathcal{O}_i}^i \mathbf{t}_i^\mathsf{T}] \right) \left( \sum_{i \in \Omega_j} \mathbb{E}[\mathbf{t}_i \mathbf{t}_i^\mathsf{T}] \right)^{-1},$$

$$\hat{\sigma}^2 = \frac{1}{\sum_{i=1}^n |\mathcal{O}_i|} \sum_{i=1}^n \left( \mathbb{E}[(\mathbf{z}_{\mathcal{O}_i}^i)^\mathsf{T} \mathbf{z}_{\mathcal{O}_i}^i] - 2\mathrm{tr}(\hat{\mathbf{W}}_{\mathcal{O}_i} \mathbb{E}[\mathbf{t}_i (\mathbf{z}_{\mathcal{O}_i}^i)^\mathsf{T}]) + \mathrm{tr}(\hat{\mathbf{W}}_{\mathcal{O}_i}^\mathsf{T} \hat{\mathbf{W}}_{\mathcal{O}_i} \mathbb{E}[\mathbf{t}_i \mathbf{t}_i^\mathsf{T}]) \right).$$

## 2.3 Approximation of the truncated normal moments

The region $g_j^{-1}(x_j^i)$ is an interval: $g_j^{-1}(x_j^i) = (a_{ij}, b_{ij})$. We may consider three cases: (1) $a_{ij}, b_{ij} \in \mathbb{R}$; (2) $a_{ij} \in \mathbb{R}, b_{ij} = \infty$; (3) $a_{ij} = -\infty, b_{ij} \in \mathbb{R}$. The computation for all cases are similar. We take the first case as an example. First we introduce a lemma for a univariate truncated normal.

**Lemma 4.** Consider a univariate random variable $z \sim \mathcal{N}(\mu, \sigma^2)$. For constants $a < b$, let $\alpha = (a - \mu)/\sigma$ and $\beta = (b - \mu)/\sigma$. Then the mean and variance of $z$ truncated to the interval $(a, b)$ are:

$$\mathrm{E}(z | a < z \le b) = \mu + \frac{\phi(\alpha) - \phi(\beta)}{\Phi(\beta) - \Phi(\alpha)} \cdot \sigma,$$

$$\mathrm{Var}(z | a < z \le b) = \left( 1 + \frac{\alpha\phi(\alpha) - \beta\phi(\beta)}{\Phi(\beta) - \Phi(\alpha)} - \left( \frac{\phi(\alpha) - \phi(\beta)}{\Phi(\beta) - \Phi(\alpha)} \right)^2 \right) \sigma^2 := h(\alpha, \beta, \sigma^2).$$

Notice conditional on known $\mathbf{z}_{\mathcal{O}_i/\{j\}}^i$, $z_j^i$ is normal with mean $\mu_{ij}$ and variance $\sigma_{ij}^2$ as

$$\mu_{ij} = \mathbf{w}_j^\mathsf{T} (\sigma^2 \mathbf{I}_k + \mathbf{W}_{\mathcal{O}_i/\{j\}}^\mathsf{T} \mathbf{W}_{\mathcal{O}_i/\{j\}})^{-1} \mathbf{W}_{\mathcal{O}_i/\{j\}} \mathbf{z}_{\mathcal{O}_i/\{j\}}^i, \tag{12}$$

$$\sigma_{ij}^2 = \sigma^2 + \sigma^2 (\sigma^2 \mathbf{I}_k + \mathbf{W}_{\mathcal{O}_i/\{j\}}^\mathsf{T} \mathbf{W}_{\mathcal{O}_i/\{j\}})^{-1} \mathbf{w}_j^\mathsf{T} \mathbf{w}_j. \tag{13}$$

Now define $\alpha_{ij} = \frac{a_{ij} - \mu_{ij}}{\sigma_{ij}}$ and $\beta_{ij} = \frac{b_{ij} - \mu_{ij}}{\sigma_{ij}}$ as in Lemma 4.

We first discuss how to estiamte $\mathrm{E}[\mathbf{z}_{\mathcal{O}_i}^i | \mathbf{x}_{\mathcal{O}_i}^i]$. Again using the law of total expectation for each $j \in \mathcal{O}_i$ by treating $\mathbf{z}_{\mathcal{O}_i/\{j\}}^i$ as known:

$$\mathrm{E}[z_j^i | \mathbf{x}_{\mathcal{O}_i}^i] = \mathrm{E}[\mathrm{E}[z_j^i | x_j^i, \mathbf{z}_{\mathcal{O}_i/\{j\}}^i] | \mathbf{x}_{\mathcal{O}_i}^i] = \mathrm{E}\left[ \mu_{ij} - \frac{\phi(\alpha_{ij}) - \phi(\beta_{ij})}{\Phi(\beta_{ij}) - \Phi(\alpha_{ij})} \sigma_{ij} | \mathbf{x}_{\mathcal{O}_i}^i \right]$$

$$= \mathbf{w}_j^\mathsf{T} (\sigma^2 \mathbf{I} + \mathbf{W}_{\mathcal{O}_i/\{j\}}^\mathsf{T} \mathbf{W}_{\mathcal{O}_i/\{j\}})^{-1} \mathbf{W}_{\mathcal{O}_i/\{j\}} \mathrm{E}[\mathbf{z}_{\mathcal{O}_i/\{j\}}^i | \mathbf{x}_{\mathcal{O}_i}^i] - \mathrm{E}\left[ \frac{\phi(\alpha_{ij}) - \phi(\beta_{ij})}{\Phi(\beta_{ij}) - \Phi(\alpha_{ij})} | \mathbf{x}_{\mathcal{O}_i}^i \right] \sigma_{ij}. \tag{14}$$

Notice $\mathrm{E}\left[ \frac{\phi(\alpha_{ij}) - \phi(\beta_{ij})}{\Phi(\beta_{ij}) - \Phi(\alpha_{ij})} | \mathbf{x}_{\mathcal{O}_i}^i \right]$ is the expectation of a nonlinear function of $\mathbf{z}_{\mathcal{O}_i/\{j\}}^i$ with respect to the conditional distribution $\mathbf{z}_{\mathcal{O}_i/\{j\}}^i | \mathbf{x}_{\mathcal{O}_i}^i$. Such expectation is intractable, thus we resort to an linear approximation:

$$\mathrm{E}\left[ \frac{\phi(\alpha_{ij}) - \phi(\beta_{ij})}{\Phi(\beta_{ij}) - \Phi(\alpha_{ij})} | \mathbf{x}_{\mathcal{O}_i}^i \right] \approx \frac{\phi(\mathrm{E}[\alpha_{ij} | \mathbf{x}_{\mathcal{O}_i}^i]) - \phi(\mathrm{E}[\beta_{ij} | \mathbf{x}_{\mathcal{O}_i}^i])}{\Phi(\mathrm{E}[\beta_{ij} | \mathbf{x}_{\mathcal{O}_i}^i]) - \Phi(\mathrm{E}[\alpha_{ij} | \mathbf{x}_{\mathcal{O}_i}^i])}. \tag{15}$$

where $\mathrm{E}[\alpha_{ij} | \mathbf{x}_{\mathcal{O}_i}^i]$ and $\mathrm{E}[\beta_{ij} | \mathbf{x}_{\mathcal{O}_i}^i]$ are linear functions of $\mathrm{E}[\mathbf{z}_{\mathcal{O}_i/\{j\}}^i | \mathbf{x}_{\mathcal{O}_i}^i]$.

Combining Eq. (14) and Eq. (15), we approximately express the $j$-th element of $\mathrm{E}[\mathbf{z}_{\mathcal{O}_i}^i | \mathbf{x}_{\mathcal{O}_i}^i]$ as a nonlinear function (including a linear part) of all other elements of $\mathrm{E}[\mathbf{z}_{\mathcal{O}_i}^i | \mathbf{x}_{\mathcal{O}_i}^i]$. Such relationship holds for all $j \in \mathcal{O}_i$, thus we have a system with $|\mathcal{O}_i|$ equations satisfied by the vector $\mathrm{E}[\mathbf{z}_{\mathcal{O}_i}^i | \mathbf{x}_{\mathcal{O}_i}^i]$.

We choose to iteratively solve this system. Concretely, to estimate $\mathrm{E}[\mathbf{z}_{\mathcal{O}_i}^i | \mathbf{x}_{\mathcal{O}_i}^i, \mathbf{W}^{(t+1)}, (\sigma^2)^{(t+1)}]$ at the $t + 1$-th EM iteration, we conduct one Jacobi iteration with $\mathrm{E}[\mathbf{z}_{\mathcal{O}_i}^i | \mathbf{x}_{\mathcal{O}_i}^i, \mathbf{W}^{(t)}, (\sigma^2)^{(t)}]$ as initial value. Surprisingly, one Jacobi iteration works well and more iterations do not bring significant improvement.

Table 1: Imputation error (NRMSE) on synthetic continuous data over 20 repetitions.

| Setting | LRGC | LRGC-Oracle |
|---|---|---|
| Low Rank | 0.347(.004)) | .330(.004) |
| High Rank | 0.517(.011) | .433(.007) |

The values of $\mu_{ij}$ and $\sigma_{ij}^2$ in Eq. (12) and Eq. (13) for all $j \in \mathcal{O}_i$ can be obtained through computing the diagonals of $(\sigma^2 \mathbf{I}_{|\mathcal{O}_i|} + \mathbf{W}_{\mathcal{O}_i} \mathbf{W}_{\mathcal{O}_i}^\mathsf{T})^{-1}$ and $(\sigma^2 \mathbf{I}_{|\mathcal{O}_i|} + \mathbf{W}_{\mathcal{O}_i} \mathbf{W}_{\mathcal{O}_i}^\mathsf{T})^{-1} \mathrm{E}[\mathbf{z}_{\mathcal{O}_i}^i | \mathbf{x}_{\mathcal{O}_i}^i, \mathbf{W}^{(t)}, (\sigma^2)^{(t)}]$, which makes the computation no more than $O(k^2 |\mathcal{O}_i|)$ for each data point at each EM iteration.

As for diagonals of $\mathrm{Cov}[\mathbf{z}_{\mathcal{O}_i}^i | \mathbf{x}_{\mathcal{O}_i}^i]$, denoted as $\mathrm{Var}[\mathbf{z}_{\mathcal{O}_i}^i | \mathbf{x}_{\mathcal{O}_i}^i] \in \mathbb{R}^{|\mathcal{O}_i|}$, using the law of total variance,

$$\mathrm{Var}[z_j^i | \mathbf{x}_{\mathcal{O}_i}^i] = \mathrm{E}[\mathrm{Var}[z_j^i | \mathbf{z}_{\mathcal{O}_i/\{j\}}^i, x_j^i] | \mathbf{x}_{\mathcal{O}_i}^i]] + \mathrm{Var}[\mathrm{E}[z_j^i | \mathbf{z}_{\mathcal{O}_i/\{j\}}^i, x_j^i] | \mathbf{x}_{\mathcal{O}_i}^i]. \tag{16}$$

In the right hand side of Eq. (16), we similarly approximate the first term, an intractable non-linear integral, as a linear term:

$$\mathrm{E}[\mathrm{Var}[z_j^i | \mathbf{z}_{\mathcal{O}_i/\{j\}}^i, x_j^i] | \mathbf{x}_{\mathcal{O}_i}^i] \approx h(\mathrm{E}[\alpha_{ij} | \mathbf{x}_{\mathcal{O}_i}^i], \mathrm{E}[\beta_{ij} | \mathbf{x}_{\mathcal{O}_i}^i], \sigma_{ij}^2). \tag{17}$$

The second term in the right hand side of Eq. (16) is also an intractable nonlinear integral. We approximate it as 0 and find it works well then linearly approximating it in practice.

Combining Eq. (16) and Eq. (17) for all $j \in \mathcal{O}_i$, we express $\mathrm{Var}[\mathbf{z}_{\mathcal{O}_i}^i | \mathbf{x}_{\mathcal{O}_i}^i]$ by nonlinear functions of $\mathrm{E}[\mathbf{z}_{\mathcal{O}_i}^i | \mathbf{x}_{\mathcal{O}_i}^i]$. Thus at the $t + 1$-th EM iteration, we first estimate $\mathrm{E}[\mathbf{z}_{\mathcal{O}_i}^i | \mathbf{x}_{\mathcal{O}_i}^i, \mathbf{W}^{(t+1)}, (\sigma^2)^{(t+1)}]$, then estimate $\mathrm{Var}[\mathbf{z}_{\mathcal{O}_i}^i | \mathbf{x}_{\mathcal{O}_i}^i]$ based on that.

## 2.4 Stopping criteria

We use the relative change of the parameter $\mathbf{W}$ as the stopping criterion. Concretely, with $\mathbf{W}_1$ from last iteration and $\mathbf{W}_2$ from current iteration, the algorithm stops if $\frac{||\mathbf{W}_1 - \mathbf{W}_2||_F^2}{||\mathbf{W}_1||_F^2}$ is smaller than the tolerance level.

## 3 Additional experiments

### 3.1 LRGC imputation under correct model

For LRGC imputation, we show the random variation of the error (due to error in the estimate of $\mathbf{z}_\mathcal{M}$) dominates the estimation error (due to errors in the estimates of the parameters $\mathbf{W}$ and $\sigma$). To do so, we compare the imputation error of LRGC imputation with estimated model parameters (LRGC) and true model parameters (LRGC-Oracle). For ordinal data, imputation requires approximating truncated normal moments, which may blur the improvement of using true model parameters. Thus we conduct the comparison on the same continuous synthetic dataset described in Section 4. The results are reported in Table 1.

Compared to LRGC, LRGC-Oracle only improves slightly (1%) over low rank data. Thus the model estimation error is dominated by the random variation of the imputation error. For high rank data, the improvement (8%) is still small compared to the gap between LRGC imputation and LRMC algorithms ($\geq 18\%$). Also notice the marginal transformation $g_j(z) = z^3$ for high rank data is not Lipschitz, so the theory presented in this paper does not bound the LRGC imputation error.

The result here indicates there is still room to improve LRGC imputation when the marginals are not Lipschitz. We leave that important work for the future.

### 3.2 Imputation error versus reliability shape with varying number of ordinal levels

We show in this section the imputation error versus reliability curve shape on ordinal data with many ordinal levels will match that on continuous data. The results here indicate the prediction power of LRGC reliability depends on the imputation task. The prediction power is larger for easier imputation

Figure 1: Imputation error on the subset of $m\%$ most reliable entries, reported over 5 repetitions.

task. In the synthetic experiments, imputing continuous data is harder than imputing ordinal data, and imputing 1-5 ordinal data is harder than imputing binary data.

We follow the synthetic experiments setting used in Section 4, but vary the number of ordinal levels to $\{5, 8, 10\}$. We adopt high SNR setting for ordinal data and low rank setting for continuous data. To make the imputation error comparable between continuous data and ordinal data, we measure the ratio of the imputation error over the $m\%$ entries to the imputation error over all missing entries. Shown in Fig. 1, the curve shape for low rank continuous data is similar to that for ordinal data with 5–8 levels. Also notice, NRMSE for continuous data involves the observed data values while MAE does not, which may cause small difference in the curve shape.

## 4 Experimental detail

Our codes are for all experiments are publictly available. [1]

**Implementation details** `softImpute` is implemented using the R package [3]. GLRM, generalized low rank model is implemented using the Julia package [9] with quadratic regularization in both factors. The implementation of GLRM is equivalent to maximum margin matrix factorization (MMMF) [6] for certain loss functions. `MMC` is implemented using the Matlab codes provided by the second author [2]. `PPCA` is implemented using the R package [7]. `MI-PCA` is implemented using the R package [5].

### 4.1 Synthetic data

To select the best value of the key tuning parameter for each method, we first run some initial experiments to determine a proper range such that the best value lies strictly inside that range.

For `LRGC` and `PPCA`, the only tuning parameter is rank. We find that a range of $6 - 14$ for continuous data (both low and high rank), and $3 - 11$ for ordinal data with 5 levels and binary data (high SNR and low SNR), suffices to ensure the best value is strictly inside the range. Notice this range is still quite small, so it is rather easy to search over.

For `softImpute`, the only tuning parameter is the penalization parameter. As suggested by the vignette of the R package [3], we first center the rows and columns of the observations using the function `biScale()` and then compute $\lambda_0$ as an upper bound on the penalization parameter using the function `lambda0()`. The penalization parameter range is set as the exponentially decaying path between $\lambda_0$ and $\lambda_0/100$ with nine points for all cases:

```
exp(seq(from=log(lam0),to=log(lam0/100),length=9)).
```

Figure 2: Imputation error over a key tuning parameter reported over 20 repetitions. The error bars are invisible. The penalization parameter $\lambda$ is plotted over the log-ratios $\log(\alpha)$ which satisfies $\lambda = \alpha\lambda_0$.

We found increasing the path length from 9 to 20 only slightly improves the performance (up to .01 across all cases) on best performance on test set.

For GLRM, there are two tuning parameters: the rank and the penalization parameter. We set the rank to be allowed maximum rank 199. We set the range of penalization parameter as we do for softImpute, with left and right endpoints that depend on the data. For GLRM-$\ell_2$ on continuous data, GLRM-BvS and GLRM-$\ell_1$ on ordinal data, we use $\lambda_0/4$ as start point and $\lambda_0/100$ as end point. For all other GLRM methods, we use $\lambda_0$ as start point and $\lambda_0/100$ as end point.

For MMC, following the authors' suggestions regarding the code, we use the following settings: (1) the number of gradient steps used to update the $Z$ matrix is 1; (2) the tolerance parameter is set as 0.01; In addition, we set the initial rank as 50, the increased rank at each step as 5, the maximum rank as 199, the maximum number of iterations as 80 and the Lipshitz constant as 10. Finally, the key tuning parameter we search over is the constant step size as suggested by the authors of [2]. The range is set as $\{3, 5, 7, \dots, 17, 19\}$.

The complete results are plotted in Fig. 2. Clearly, LRGC and PPCA does not overfit even for high ranks, across all settings. We also provide the runtime for each method at the best tuning parameter in Table 2. Notice our current implementation is written entirely in R, and thus further acceleration is possible.

## 4.2 MovieLens 1M

The dataset can be found at `https://grouplens.org/datasets/movielens/1m/`. Similar to the synthetic experiments, we choose the tuning parameter for each method on a proper range determined through some initial experiments. For LRGC, we choose the rank from $\{8, 10, 12, 14\}$ to be 10. With $\lambda_0$ calculated as for the synthetic data, for softImpute, we select the penalization parameter from $\{\frac{\lambda_0}{2}, \frac{\lambda_0}{4}, \frac{\lambda_0}{6}, \frac{\lambda_0}{8}\}$ to be $\frac{\lambda_0}{4}$; for GLRM-BvS, we set the rank as 200 and select the penalization parameter from $\{\frac{\lambda_0}{10}, \frac{\lambda_0}{12}, \frac{\lambda_0}{14}, \frac{\lambda_0}{16}, \frac{\lambda_0}{18}\}$ to be $\frac{\lambda_0}{14}$.

We report detailed results in Table 3. We see that all the models perform quite similarly on this large dataset. In other words, the gain from carefully modeling the marginal distributions (using a LRGC) is insignificant. This phenomenon is perhaps unsurprising given that sufficiently large data matrices from a large class of generative models are approximately low rank [8].

Table 2: Run time (in seconds) for synthetic data at the best tuning parameter; mean (variance) reported over 20 repetitions.

| Continuous | LRGC | PPCA | softImpute | GLRM-$\ell_2$ | MMC |
|---|---|---|---|---|---|
| Low Rank | 5.7(0.2) | 2.9(0.4) | 0.7(0.0) | 3.3(1.0) | 457.9(10.4) |
| High Rank | 6.5(0.3) | 0.3(0.1) | 1.1(0.2) | 7.6(2.0) | 554.4(32.0) |
| 1-5 ordinal | LRGC | PPCA | softImpute | GLRM-BvS | GLRM-$\ell_1$ |
| High SNR | 27.2(0.7) | 1.0(0.2) | 1.2(0.1) | 19.2(1.5) | 17.4(1.2) |
| Low SNR | 19.8(0.8) | 0.3(0.1) | 1.3(0.0) | 17.4(1.5) | 17.0(1.4) |
| Binary | LRGC | PPCA | softImpute | GLRM-hinge | GLRM-logistic |
| High SNR | 66.7(3.0) | 0.9(0.5) | 1.4(0.3) | 3.8(0.3) | 4.4(0.6) |
| Low SNR | 52.0(4.4) | 0.3(0.1) | 1.5(0.2) | 3.4(0.4) | 3.3(0.4) |

Table 3: Imputation error for MovieLens 1M over 5 repetition. Run time is measures in minutes.

| Algorithm | MAE | RMSE | Run time |
|---|---|---|---|
| LRGC | **0.619**(**.002**) | 0.910(.003) | 38(1) |
| softImpute | 0.629(.003) | **0.905**(**.003**) | 93(2) |
| GLRM-BvS | 0.633(.002) | 0.921(.002) | 25(1) |

## Footnotes

[1]https://github.com/yuxuanzhao2295/Matrix-Completion-with-Quantified-Uncertainty-through-Low-Rank-Gaussian-Copula