[Reviews · NeurIPS 2020]

Review 1

Summary and Contributions: This paper studies the multiple imputation problem on the basis of the low-rank Gaussian copula model (I believe it is more suitable to use "multiple imputation" rather than "matrix completion" for this paper). The imputation errors are evaluated both theoretically and empirically.

Strengths: The proposed techniques are reasonable, the theoretical results seem solid, and the experimental results are competitive.

Weaknesses: 1. The actual contribution of the paper is unclear to me. The authors claim to invent a matrix completion method that owns the rare ability of quantifying uncertainty; this is unconvincing. In fact, there are two different perspectives for studying the missing data problem: matrix/tensor completion and multiple imputation. Each perspective has its own strenghts and application scope. In matrix completion, which aims to restore a signal presented in matrix form from a sampling of its entities, uncertainty exists only in the presence of observation noise and is therefore rarely considered by the community; this is exactly the motivation of ref[5]. As for multiple imputation, which treats the data as being generated by some probabilistic model, quantified uncertainty is indeed a very common, trivial property. It seems to me that this paper is actually about multiple imputation but written from the perspective of matrix completion. But, from the view point of matrix completion, many important attributes such as the identifiability conditions are entirely missing from the paper. In summary, I cannot be sure about the contributions of the paper, unless the proposed techniques are clearly compared with multiple imputation methods, both methodologically and experimentally. 2. As for the novelty of the proposed techniques, I am not convinced either. Gaussian copula model has been widely used in multiple imputation, e.g., Multiple Imputation Using Gaussian Copulas. The combination with low-rank factorization, i.e., the proposed low-rank Gaussian copula (LRGC), seems straightforward.

Correctness: Some claims seem inappropriate, see above. I have no complain about the empirical methodology.

Clarity: The language usage is good. But the movations are unclear, see above.

Relation to Prior Work: No. see above.

Reproducibility: Yes

Additional Feedback:


Review 2

Summary and Contributions: The paper focuses on uncertainty quantification in matrix completion problems. The authors propose a probabilistic framework based on low-rank copula, which allows provable uncertainty quanfitication for imputation of continuous and categorical data. They introduce a scalable algorithm adapted to high-dimensional problems, and an empirical study on large-scale recommender system data which demonstrates the competitive results of the method compared to state of the art techniques.

Strengths: The paper proposes a solution to an important problem in missing values imputation: the quantification of uncertainty. Most existing techniques (such as multiple imputation) are very costly, and to my knowledge this is the first paper to propose a scalable method adapted to data sets with millions if observations. The article is very thorough, with an innovative framework for quantification of uncertainty in imputation, theoretical results, an efficient algorithm and implementation, and finally an extensive simulation study on realistic real-world problem.

Weaknesses: In my opinion, one weakness of the paper is the discussion of related work, which I did not find really complete. See details in the relation to prior work section.

Correctness: The theoretical claims and empirical methodology are correct.

Clarity: The paper is very clear and well written, I enjoyed reading it

Relation to Prior Work: In my opinion a few related fields are not discussed enough: - multiple imputation methods which allow uncertainty quantification including for continuous, categorical or mixed data (e.g package R missMDA, or methods based on generative models http://proceedings.mlr.press/v97/mattei19a/mattei19a.pdf) - papers discussing uncertainty quantification in matrix completion problems, which are not referenced either (e.g. https://arxiv.org/abs/1704.02760)

Reproducibility: Yes

Additional Feedback: I have read the author feedback, and appreciate the fact that the authors have included the reference I pointed out. I thus still think this is a good paper and advocate for acceptance.


Review 3

Summary and Contributions: The paper introduces a Gaussian Copula model with correlations being modeled as low-rank by a probabilistic factor model. Inference is done using EM. The model is used for matrix completion and quantify uncertainty of its predictions. It outperforms other methods on a synthetic task that match its modeling assumptions.

Strengths: - Thorough formal arguments and good paper structure. - Well written.

Weaknesses: - The paper could provide a higher-level intuition of many of the technical steps. This is a missed opportunity. - The choice of competing methods is limited. For instance use MICE or even a supervised approach like DataWig (https://github.com/awslabs/datawig) that can also quantify uncertainty. - Hyper-parameter selection on competing methods is unclear. This weakens the experimental evaluation. - Evaluation on real-world data is scarce. The actual performance benefit seems rather marginal. This good be strengthened with more real-world experiments. - The extent to which a model's uncertainty measure predicts the true prediction error is often referred to as "calibration", see for instance Guo et al., 2017. (http://proceedings.mlr.press/v70/guo17a/guo17a.pdf)

Correctness: Yes

Clarity: The paper is well written but heaviness on mathematical notation could be reduced. The paper mixes math and natural language in an unusual way. E.g. "let [p] = {1,...p} for p \in Z", where the predicate of the sentence is given as mathematical symbol.

Relation to Prior Work: - Discussion of prior work is insufficient. There exist many probabilistic factor models, e.g. or instance [A] and [B]. The authors state that none of the prior works they cite can quantify uncertainty. This is not true (e.g. for author's reference [3]. - Not all Bayesian approaches rely on "expensive MCMC" some are made scalable (e.g. [B]) others use can work with mini-batched variational inference. [A] Mohamed, S. (2011). Generalised Bayesian matrix factorisation models (Doctoral thesis). https://doi.org/10.17863/CAM.13995 [B] Tammo Rukat, Chris C. Holmes, Michalis K. Titsias, Christopher Yau; Bayesian Boolean Matrix Factorisation. Proceedings of the 34th International Conference on Machine Learning, PMLR 70:2969-2978, 2017. Experiments - "Our experiments show that LRGC imputations are accurate" -> This is meaningless without context. - "because imputing continuous data is in general harder than imputing ordinal data." In the given generality this is a meaningless statement.

Reproducibility: Yes

Additional Feedback: - Abstract: "Almost none [imputation algorithm ]can estimate uncertainty of its imputations." This is not true, there is a plethora of approaches that offer direct or indirect ways of quantifying uncertainty. Introduction - "each missing entry has a deterministic distribution conditional on the observations". What does "deterministic" mean here? - "Although our proposed model has a low rank structure, it greatly differs from LRMC in that the observations are assumed to be generated from, but not equal to, a real-valued low rank matrix." Unclear what this means. Many low rank matrix completion approaches have an underlying generative model. Experiments - It is interesting to see that MMMF performs worse on the items that LRGC takes as most reliable. This needs some explanation. If this is a real effect, than combining the two models would be beneficial


Review 4

Summary and Contributions: This paper proposes a matrix completion method equipped with a tractable uncertainty measure (termed the reliability) based on a low-rank latent factor modeling using Gaussian copulas (LRGC). The proposal for the reliability measure is to use the sample analog of certain probability bounds, which are derived assuming access to the underlying parameters, namely a confidence interval or a lower bound of the probability of making a correct prediction, depending on the data type of the matrix. The experiments using synthetic or Movielens 1M datasets demonstrate that the proposed model compares favorably to some baseline imputation methods and that the proposed reliability measure can rank the missing indices well in the order of the estimation errors.

Strengths: [Soundness] - The proposed reliability measure has decent theoretical motivation. - The effectiveness of the proposed method is demonstrated well through the experiments. - The presentation is clear, well-organized, and easy to follow. [Significance] Since matrix completion has been pervasive in machine learning as a generic tool, e.g., in semi-supervised learning and collaborative filtering, proposing a method equipped with both a tractable uncertainty measure and a sufficient representation power can be considered an important contribution to the field, in the absence of such a method. [Novelty] The proposed reliability measure seems non-trivial, contributing to the originality of the paper. [Relevance] Matrix completion is one of the pervasive techniques in machine learning with a wide range of applications, hence its development can be of interest to a general audience in the field.

Weaknesses: [Soundness] - The theoretical analysis has the weakness that it does not take into account the estimation error of the proposed estimation procedure. - The experimental validation mainly relies on synthetic data, and the evaluation using real-world datasets is not comprehensive. It would be desirable to evaluate the performance with a few more benchmark datasets.

Correctness: To the best of my understanding, the claims and the methods are sound and correct. The experiments seem to substantiate the claims well.

Clarity: The paper is generally very well written and easy to follow. Here, I have some suggestions for further improvements. [Clarity: introduction] - The introduction states (in "Our contribution" paragraph) "We characterize how the mean squared error (MSE) of our imputations depends on the SNR." Since the theoretical analysis disregards the estimation error (Section 3.4), I believe this statement is a little misleading although I do not consider it as a ground for rejection. - In the abstract, where "state-of-the-art imputation accuracy across a wide range of datasets" is stated, it may confuse the readers into thinking that the experiments involve multiple real-world benchmark datasets instead of synthetic ones. I would like the authors to clarify in this part that the "datasets" also involve synthetic data. [Clarity: technical] - The relation of Condition (2) in Definition 2 and $WW^\top + \sigma^2 I_p$ being a correlation matrix is unclear. Specifically, in order for this matrix to be a correlation matrix, the off-diagonal entries need to be in the range of [-1, 1]. Although Condition (2) does imply this off-diagonal condition, the reason why Condition (2) suffices for the matrix to be a correlation matrix should be made more explicit in the text. - Lines 104-105 and Definition 2 both state a definition of LRGC, and the relation is unclear. Please merge Lines 104-105 into Definition 2 or explain in the text how they relate to each other (e.g., they are equivalent). - Definition 3 states "Suppose x \sim LRGC(...)." However, this seems to be an unnecessary complication. Most importantly, this statement makes the notation $W$ to be the ground-truth parameter although the imputation value $\hat x_{\mathcal{M}}$ also depends on $W$, which can differ from the ground-truth matrix (an estimator is used instead of the ground-truth in the proposed method). - Similarly, the quantities such as $\hat x_{\mathcal{M}}$ in Definition 3 or Equation (5-6) depend on the ground-truth parameters $W, \sigma^2, g$, thus they cannot be computed in reality. To make it explicit what is the actual proposed estimators, it would be desirable to distinguish the proposed estimators from these imaginary quantities. - Line 274: $Z^3$ is normally used to express the matrix multiplication, so please clarify that the cubic is entry-wise (if this is not matrix multiplication). [Other: typographical errors] - Line 47: and quantifying -> and quantify - Line 70: $\mathbb{Z}$ -> $\mathbb{N}$ - Line 136: $x_\mathcal{M}$ probably should be $\mathcal{M}$? - Line 160: (in the denominator) $\|...\|_2$ -> $\|...\|_F$ - Lines 167, 216, etc.: should $/(i,j)$ be $\setminus (i, j)$? - Lines 167-168: Please distinguish the missing entry $(i, j)$ and the temporary indices $(i, j)$ used to describe $D_\alpha$, e.g., by changing the latter to $(i', j')$. - Lines 187: "relax to" -> "relax it to" - Line 237: "aref" -> "are" - Line 261: "second author [13]" -> "second author of [13]" (otherwise, it could be read as if the second author of the submitted paper is one of the authors of [13]) [Other: terminology] - Line 152: "the marginal $g_j$" may be inappropriate since this function is not a marginal distribution/density function (or they are typically the quantile function). [Others] - Line 184-185: "is chosen to ensure finite output" -> Please clarify what this is referring to (I read it as the finite output of $g_j^{-1}$). - Line 191: "consistent" -> Please clarify what consistency (with respect to what limit) this is referring to (I read it as the consistency w.r.t. $n -> \infty$). - Line 203: $\mathbb{E}$ -> The notation may be confusing because $\mathbb{E}$ is often generically used for unconditional expectation. It would be desirable to distinguish the notation, e.g., by adding a few super-/sub-scripts such as $\mathbb{E}^i_{| X_\mathcal{O}}$. - Lines 237-238: "singular values of $W$ as $\lambda_1(W)$ and $\lambda_k(W)$" -> The notation $W$ has already been used for the ground-truth parameter. Please distinguish the temporary variable, e.g., by using $W'$ for the temporary variable. - Line 283: "entries with higher reliability (smaller $m$)" -> This bracket comment seems unnecessary because $m$ is a parameter of the evaluation metric, so I believe smaller $m$ does not have a direct connection to entries with higher reliability. - Table 1: I believe the numbers are the averages over the 20 repetitions and the numbers in the brackets are the standard errors. If so, please clarify that in the caption.

Relation to Prior Work: The introduction discussed the relation to previous contributions, where the originality of the paper is clarified.

Reproducibility: Yes

Additional Feedback: [Questions] - Lines 158-159: "our measure of reliability is designed so that reliable imputations have a low normalized root mean squared error [...]" Is this a conjecture or a theoretically justified statement? - Lines 169-173: How does the proposed uncertainty measure compare to a simpler alternative of using $(D_\alpha)^i_j / (\hat X)^i_j$? - Lines 253-257: Why is the evaluation of the reliability based on a metric that is aggregated over the top-m% entries? It seems possible to more naively plot the relation of the imputation error and the uncertainty metrics or to evaluate the rank correlation between the imputation error and the uncertainty metric. [High-level assessments] - Is it possible to naturally extend the approach to mixed data types, e.g., some columns being continuous and others being ordinal? Although such a situation is unlikely in collaborative filtering where columns tend to be items, I believe the method may find further use cases if such mixed data can be handled in an integrated manner. - Line 229: "extremely challenging" -> Why? It would be desirable to note what makes this challenging here, if possible. - Theorem 3: Can you also possibly provide a model misspecification error bound that theoretically guarantees the performance even when the ground-truth data arise from other copulas than LRGC? ================ Update after reading the author feedback: I have read the author response. The response reasonably answered some of my questions, and I believe the authors will update the manuscript to clarify some of the points that appeared in the reviews/response. My score remains the same (with a positive inclination towards advising an acceptance) as the response did not resolve my concerns but it was mostly for clarification or answering to suggestions for improvements.

[Author Response · NeurIPS 2020]

We thank the reviewers for providing useful feedback. To address a shared comment, we first explain our contribution in uncertainty quantification, compared to multiple imputation (MI) methods. While previous MI work can quantify sample variance, few papers explicitly explore the issue of *calibration*: does MI sample variance predict imputation accuracy? Our paper does address calibration: imputation accuracy correlates with our uncertainty metric. Moreover, our method allows for fast large-scale computation and uses only a single easy-to-choose parameter: rank. In contrast, Bayesian MI methods are less user-friendly: they are sensitive to the choice of prior and the selection of (often many) tuning parameters. We compare our method with one of the fastest MI methods, MIPCA (Josse et al. 2011), on synthetic data (here, Figure 1), which shows the MI sample variance does not predict imputation accuracy well. Worse, on the Movielens 1M data, even a single imputation of MIPCA cannot finish in 3h (and 20 imputations are usually used to quantify uncertainty); our method takes 38m. The author of the famous MI method MICE warns against its use on data sets with many columns due to both speed and quality consideration (Van Buuren 2018 Chapter 9.1). On our simulated low rank continuous matrix ($500 \times 200$), a single imputation of MICE ran 4m, while our method only ran 7s. Hence our new proposed uncertainty measure outperforms competitors in both accuracy and speed.

**Reviewer #2** asks us to clarify the relation between our work, multiple imputation (MI), and matrix completion (MC). We use the term MC to mean the task of imputing missing values in a tabular data set (a broader definition than R2 uses), and MI to mean methods that provide multiple imputed values for each missing entry. While many so-called MC methods assume a deterministic signal (and so require deterministic assumptions e.g. incoherence), probabilistic models including MI are also widely

Figure 1: Imputation error (NRMSE for continuous and MAE for ordinal) on the subset of $m\%$ entries for which method's associated uncertainty metric indicates highest reliability. For MIPCA, we use 20 imputations; low sample variance corresponds to high reliability.

used for tabular imputation (and so require probabilistic assumptions e.g. subGaussianity, as in our Cor. 1). Our method is a new probabilistic single imputation approach for tabular imputation. It can also be used for MI if desired.

**Reviewer #2** doubts the novelty of our methods. This paper presents the first imputation method using the Gaussian copula that scales to the recommendation system setting. While there exist methods for MI using the Gaussian copula, they all use MCMC and require large computation to achieve desirable imputation accuracy. Zhao & Udell (2019) proposed a frequentist algorithm for imputation using Gaussian copula, which is faster than the MCMC algorithms but still scales as the cube of the number of columns. Our methodological innovation is to use low rank factorization to reduce the complexity, which yields new probabilistic error guarantees (Thm 3).

**Response to Reviewer #3** We appreciate the positive comments and have added the suggested references. Unfortunately, the honest confidence intervals proposed by Carpentier et al. (2017) depend on (possibly huge) hidden constants.

**Reviewer #4** suggests comparing to DataWig (and MICE, see above). Datawig is an imputation method aimed at non-numerical (e.g. text) missing data without a direct way to quantify uncertainty, so we cannot see how to compare.

**Reviewer #4** asks how we selected hyper-parameters in experiments. As detailed in the supplement, we selected hyper-parameters for all methods using grid-search. We also show our method is not sensitive to its hyper-parameter.

We thank **Reviewer #4** for the suggested references and will add them in the revision, and for the suggestion to call our task (of predicting imputation accuracy) "calibration". But it seems these references do not address the task of calibration for ordinal or real-valued matrices. Our reference [3] does not discuss uncertainty quantification. The reference [A] does not discuss calibration. The reference [B] can calibrate imputation accuracy for Boolean matrices but does not generalize to ordinal or real-valued matrices. We will clarify that "expensive MCMC" applies only to our references [28, 6, 25, 11], the Bayesian approaches most similar to ours, rather than to all Bayesian imputation.

**Minor comments from Reviewer #4**: (1) In "…deterministic distribution…", you're right that we meant "closed form", not "deterministic". (2) We agree that many low rank models have an underlying generative model, and addressed this point immediately following the sentence quoted by the reviewer. (3) The reviewer's suggestion ("Experiments") results from a misinterpretation of Figure 1 of the paper. The figure shows that MMMF performs worse on the items that MMMF (not LRGC) takes as most reliable. In other words, the reliability metric for MMMF is *negatively* correlated with true reliability. This result indicates sample variance does not correlate with imputation uncertainty.

**Response to Reviewer #5.** We appreciate the positive comments and will improve the presentation clarity. (1) Analyzing the estimation error is challenging because EM algorithms are only guaranteed to converge to a local maximizer rather than the global maximizer and our objective likelihood function is nonconvex. (2) "Our measure...low NRMSE" is a design principle rather than theoretical property. (3) Empirically, we found the measure using $D_\alpha(i,j)/|\hat{x}^i_j|$ correlates less well with imputation error. (4) We use top-m% entries because the imputation error on a single point yields noisy plots. (5) Our imputation and uncertainty quantification methods extend to mixed data quite easily.

[Meta-Review · NeurIPS 2020]

This paper proposed low-rank Gaussian copula for matrix completion which enables uncertainty quantification. The reviews are divided after author response and rounds of discussions. The positive reviews focus on the theoretical characterization of confidence intervals for imputation and the scalability of the proposed model compared to other similar approaches. On the other hand, a handful of reviews pointed out the novelty of the approach is limited and it is not completely agreed upon whether this paper focuses on matrix imputation or matrix completion, which has different theoretical characterization. Another common negative point is that the related work is not thoroughly discussed. Overall, I think the pros slightly overweight the cons, especially if the authors can improve the discussion around the related work in the revision. However, I wouldn't mind if the decision is overturned.